# Long-term transverse imaging of the hippocampus with glass microperiscopes

William T Redman[1], Nora S Wolcott[2], Luca Montelisciani[3], Gabriel Luna[4],
Tyler D Marks[4], Kevin K Sit[5], Che-Hang Yu[6], Spencer Smith[4,6], Michael J Goard[2,4,5]*

[1]Interdepartmental Graduate Program in Dynamical Neuroscience, University of California, Santa Barbara, United States; [2]Department of Molecular, Cellular, and Developmental Biology, University of California, Santa Barbara, United States; [3]Cognitive and Systems Neuroscience Group, University of Amsterdam, Amsterdam, Netherlands; [4]Neuroscience Research Institute, University of California, Santa Barbara, United States; [5]Department of Psychological and Brain Sciences, University of California, Santa Barbara, United States; [6]Department of Electrical and Computer Engineering, University of California, Santa Barbara, Santa Barbara, United States

**Abstract** The hippocampus consists of a stereotyped neuronal circuit repeated along the septal-temporal axis. This transverse circuit contains distinct subfields with stereotyped connectivity that support crucial cognitive processes, including episodic and spatial memory. However, comprehensive measurements across the transverse hippocampal circuit in vivo are intractable with existing techniques. Here, we developed an approach for two-photon imaging of the transverse hippocampal plane in awake mice via implanted glass microperiscopes, allowing optical access to the major hippocampal subfields and to the dendritic arbor of pyramidal neurons. Using this approach, we tracked dendritic morphological dynamics on CA1 apical dendrites and characterized spine turnover. We then used calcium imaging to quantify the prevalence of place and speed cells across subfields. Finally, we measured the anatomical distribution of spatial information, finding a non-uniform distribution of spatial selectivity along the DG-to-CA1 axis. This approach extends the existing toolbox for structural and functional measurements of hippocampal circuitry.

*For correspondence:
michael.goard@lifesci.ucsb.edu

Competing interest: The authors declare that no competing interests exist.

## Editor's evaluation

This paper presents new optical technologies that allow the investigation of all stages of the tri-synaptic hippocampal circuit during behavior within an individual animal. The approach is a major methodological advance and the authors use it to confirm differences in spatial selectivity of place cells across hippocampal subregions. The paper will be of interest to the large number of neuroscientists who study the hippocampal circuit, and more broadly to those interested in methods to enable high-resolution in vivo imaging across multiple depths in the brain.

## Introduction

The hippocampus is critical for episodic and spatial memory formation (*O'Keefe and Nadel, 1978*; *Squire, 1992*; *Tonegawa et al., 2015*), but the neural computations underlying these functions are not well understood. The trisynaptic circuit linking entorhinal cortex (EC) to dentate gyrus (DG), DG to CA3, and CA3 to CA1 is believed to endow the hippocampus with its functional capabilities. Since the circuit was first described in the anatomical studies of Ramon y Cajal over a century ago (*Ramon, 1911*), considerable work has focused on each of the major hippocampal subfields (CA1-3 and DG) to identify their roles in hippocampal processing. The resulting body of literature has indicated that the

subfields have related, but distinct roles in pattern separation and completion (*Gilbert et al., 2001*; *Lee et al., 2004*; *Leutgeb et al., 2004*; *Leutgeb et al., 2005*; *Leutgeb et al., 2007*; *McHugh et al., 2007*; *Nakashiba et al., 2012*; *Kheirbek et al., 2013*; *Neunuebel and Knierim, 2014*; *Rennó-Costa et al., 2014*), response to novelty (*Frank et al., 2004*; *Karlsson and Frank, 2008*; *Kemere et al., 2013*; *Larkin et al., 2014*; *Dong et al., 2021*), and the encoding of social variables (*Hitti and Siegelbaum, 2014*; *Okuyama et al., 2016*; *Meira et al., 2018*). Additionally, there appear to be differences between the subfields in the stability of their place fields (*Dong et al., 2021*; *Mankin et al., 2012*; *Hainmueller and Bartos, 2018*). Although this work has increased our understanding of each of the subfields individually, it is not yet clear how neuronal activity is coordinated across the hippocampus.

This lack of knowledge comes, in part, from the technological limitations that prevent the recording of neuronal populations across hippocampal subfields in the same animal. Historically, electrophysiology has been the principal tool used to study the hippocampus. Electrophysiological recordings have the advantage of high temporal resolution and can directly measure spiking, but they are typically limited to small numbers of neurons in particular subfields. Additionally, localization of recorded neurons within the hippocampus is approximate, requiring post hoc histological analysis to estimate the position of the electrode tracks, with the distances between the recorded neurons and the electrode sites being poorly defined. In recent years, calcium imaging approaches (i.e. single-photon mini-endoscopes and two-photon [2P] microscopy) have been used to record hippocampal activity, allowing for the simultaneous measurement of large numbers of neurons with known spatial relationships (*Hainmueller and Bartos, 2018*; *Dombeck et al., 2010*; *Ghosh et al., 2011*; *Ziv et al., 2013*; *Cai et al., 2016*; *Sheintuch et al., 2017*). However, these approaches require aspiration of the overlying neocortex and are generally restricted to a single subfield for each animal.

Taken together, current experimental techniques are limited in their ability to: (1) record response dynamics and coordination across the hippocampus, (2) identify and distinguish between different neural subtypes, (3) allow for the chronic recording of cells across subfields, and (4) resolve key cellular structures, such as apical dendrites.

To address these challenges, we have developed a procedure for transverse imaging of the trisynaptic hippocampal circuit using chronically implanted glass microperiscopes. As has been found with previous studies using implanted microprisms in cortex (*Chia and Levene, 2009*; *Andermann et al., 2013*; *Low et al., 2014*), the neural tissue remained intact and healthy for prolonged periods of time (up to 10 months), and both dendritic structure and calcium activity could be repeatedly measured in awake mice. Optical modeling and point spread function (PSF) measurements showed that axial resolution is decreased compared to traditional cranial windows, but is sufficient to image individual apical dendritic spines in hippocampal neurons several millimeters below the pial surface. Using this approach, we quantified spine turnover in CA1 apical dendrites across days. We then measured functional responses from CA1, CA3, and DG in head-fixed mice as they explored a floating carbon fiber arena (*Kislin et al., 2014*; *Go et al., 2021*). We found neurons in all regions whose activity met criteria to be considered place cells (PCs) and speed cells (SCs). Further, we found a non-uniform distribution of spatial information across the extent of the DG-to-CA1 axis, supporting findings from earlier electrophysiological studies that found similar heterogeneity (*Lee et al., 2015*; *Lu et al., 2015*; *Mankin et al., 2015*). Taken together, this approach adds to the existing neurophysiological toolkit and will enable chronic structural and functional measurements across the entire transverse hippocampal circuit.

## Results

### Optical access to the transverse hippocampus using implanted microperiscopes

To image the transverse hippocampal circuit using 2P imaging, we developed a surgical procedure for chronically implanting a glass microperiscope into the septal (dorsomedial) end of the mouse hippocampus (*Figure 1A*; *Figure 1—figure supplement 1*; see Materials and methods). For imaging CA1 only, we used a $1 \times 1 \times 2$ mm$^3$ microperiscope (v1$_{CA1}$; *Figure 1B*, left), and for imaging the entire transverse hippocampus (CA1-CA3, DG), we used a $1.5 \times 1.5 \times 2.5$ mm$^3$ microperiscope (v2$_{HPC}$; *Figure 1B*, right). The microperiscope hypotenuse was coated with enhanced aluminum in order to reflect the imaging plane orthogonally onto the transverse plane (*Figure 1C and D*). To insert the

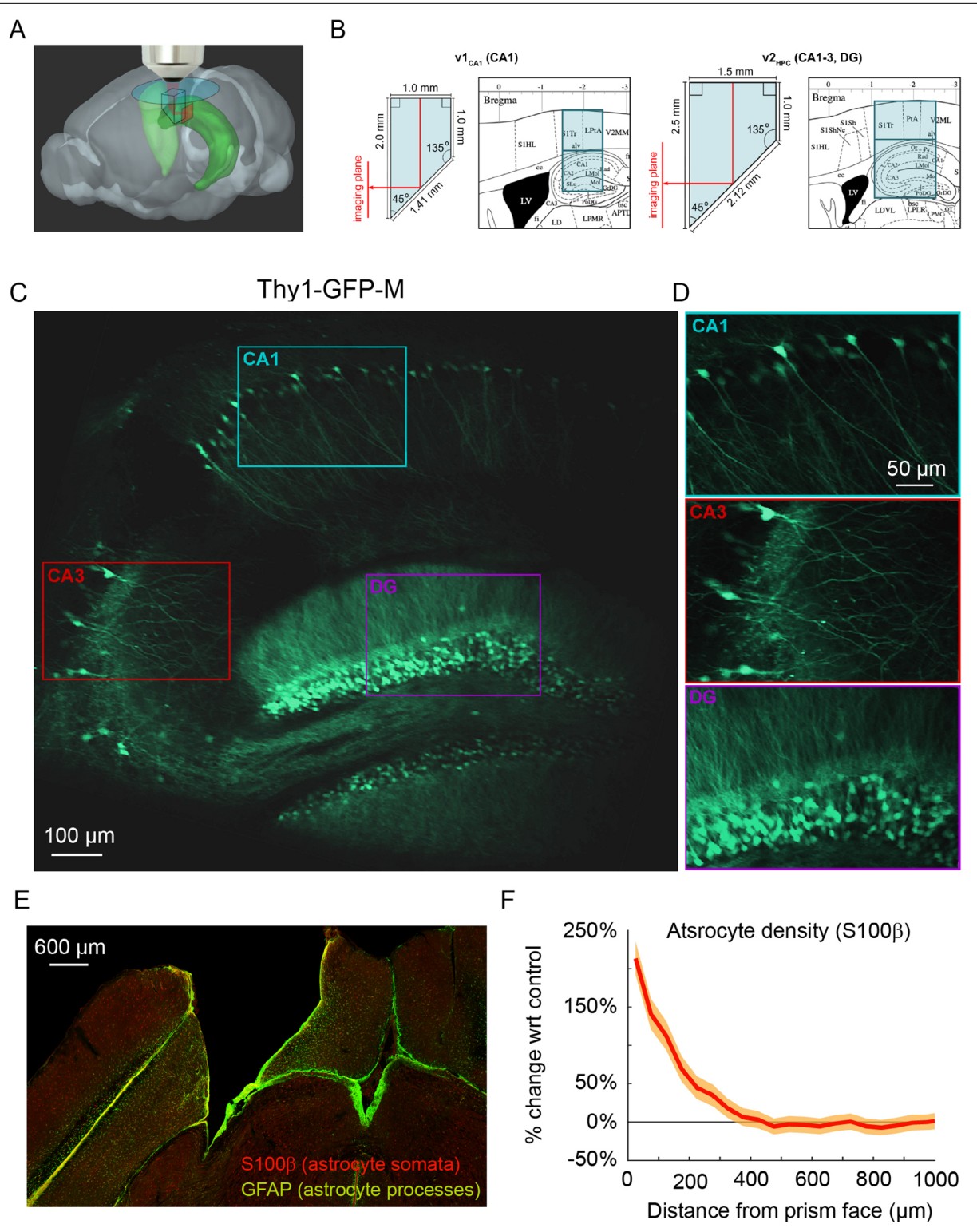

**Figure 1.** Implanted microperiscopes allow imaging of the hippocampal transverse plane. (**A**) Three-dimensional schematic (*Wang et al., 2020*) illustrating microperiscope implantation and light path for hippocampal imaging. (**B**) Schematics (*Paxinos and Franklin, 2001*) showing the imaging plane location of v1$_{CA1}$ (1 mm imaging plane, 2 mm total length) and v2$_{HPC}$ (1.5 mm imaging plane, 2.5 mm total length) microperiscopes. (**C**) Tiled average projection of the transverse imaging plane using the v2$_{HPC}$ microperiscope implant in a Thy1-GFP-M transgenic mouse. Scale bar = 100 µm. (**D**) Enlarged images of hippocampal subfields (CA1, CA3, DG) corresponding to the rectangles in (**C**). Scale bar = 50 µm. (**E**) Example histological section stained for astrocyte cell bodies (S100β, red) and processes (GFAP, green). Scale bar = 600 µm. (**F**) Quantification of astrocyte density as a

*Figure 1 continued on next page*

*Figure 1 continued*

function of distance from the microperiscope face, normalized to the density in the unimplanted contralateral hemisphere (*n*=2 mice; 288 and 316 days post-implant; mean ± bootstrapped s.e.m.).

The online version of this article includes the following figure supplement(s) for figure 1:

**Figure supplement 1.** Overview of microperiscope implantation surgery.

**Figure supplement 2.** Additional visualization of astrocyte immunohistochemistry.

microperiscope, we made an incision through the dura and tissue (*Figure 1—figure supplement 1A*), then lowered the tip of the microperiscope into the incision (*Figure 1—figure supplement 1B*), pushing the cortical tissue medially (*Figure 1—figure supplement 1C*). Although this approach eliminated the need for the aspiration of cortical tissue typically performed prior to hippocampal imaging (*Dombeck et al., 2010*), it nonetheless results in severed connections and compressed tissue medial to the implant. Since the septal end of the hippocampus was affected by the implant, we used immunohistochemistry to quantify the effect of microperiscope implantation on astrocyte proliferation, as a function of distance from the imaging face of the microperiscope (*Figure 1E*; *Figure 1—figure supplement 2*). Similar to previous research using microprism implants (*Andermann et al., 2013*), we found an increase in the prevalence of astrocytes close to the microperiscope face, but the density decreased with distance and was indistinguishable from the control hemisphere beyond 300 µm from the microperiscope face (*Figure 1F*).

## Characterizing optical properties of microperiscope

Use of the microperiscope requires imaging through several millimeters of glass, which could cause beam clipping or optical aberrations, potentially resulting in decreased signal intensity and optical resolution. However, the extent to which imaging through glass prisms affects optical signal quality is not well understood. To determine how our approach affects 2P image properties, we modeled the expected optical properties and compared it to the experimentally determined signal intensity and PSF measurements using fluorescent microspheres (*Figure 2A*). To characterize signal intensity, we measured the minimum laser power necessary for saturation at the center of the microsphere, holding other imaging parameters constant (*Figure 2B*). We found that signal intensity imaged through the microperiscope was reduced compared to a standard coverslip (v1$_{CA1}$ microperiscope: 59.0% ± 6.6%; v2$_{HPC}$ microperiscope: 67.2% ± 8.5%; mean ±s.d.; *Figure 2C*). To measure resolution, we imaged small diameter (0.2 µm) fluorescent microspheres and determined the PSF, measured as the full width at half maximum (FWHM) of the lateral and axial microsphere profiles. Compared to a standard cranial window using a 0.15 mm coverslip, we found that the FWHM was higher for both lateral (coverslip: 0.6 µm; v1$_{CA1}$ microperiscope: 1.0 µm; v2$_{HPC}$ microperiscope: 0.8 µm; *Figure 2D–G*) and axial (coverslip: 3.4 µm; v1$_{CA1}$ microperiscope: 11.7 µm; v2$_{HPC}$ microperiscope: 9.5 µm; *Figure 2D–G*) dimensions. Optical modeling indicated that the decrease in resolution is predominantly due to clipping of the excitation beam, resulting in a reduction of the functional numerical aperture of the imaging system (theoretical axial PSF FWHM of v1$_{CA1}$ microperiscope: 10.9 µm; v2$_{HPC}$ microperiscope: 7.7 µm; see Materials and methods) rather than optical aberrations. As a result, use of adaptive optics did not significantly improve the axial resolution, though adaptive optics did improve the signal intensity by 40–80% (data not shown). We next measured whether the optical properties changed across the microperiscope imaging plane, measuring the lateral and axial microsphere FWHM across the horizontal extent of the microperiscope (*Figure 2H*). We found that the resolution was mostly uniform, with no significant differences observed in lateral or axial resolution as a function of horizontal position (*Figure 2I*; lateral: $F(14, 593)$=0.53, p=0.91; axial: $F(14, 593)$=0.91, p=0.55; one-way ANOVA). Finally, we measured resolution as a function of distance from the face of the microperiscope (*Figure 2J*). The lateral FWHM did not change as a function of distance from the microperiscope face (*Figure 2K*; $F(19, 109)$=0.53, p=0.26; one-way ANOVA). The axial FWHM slightly increased with greater distances from the microperiscope face (*Figure 2K*; $F(19, 109)$=2.51, p=0.002; one-way ANOVA), though none of the individual positions were significantly different when corrected for multiple comparisons (p>0.05 for all positions, Tukey-Kramer post hoc test).

Taken together, these results indicate that imaging through the microperiscope lowers signal intensity and axial resolution. Despite these effects, the quality is still sufficient to image individual HPC

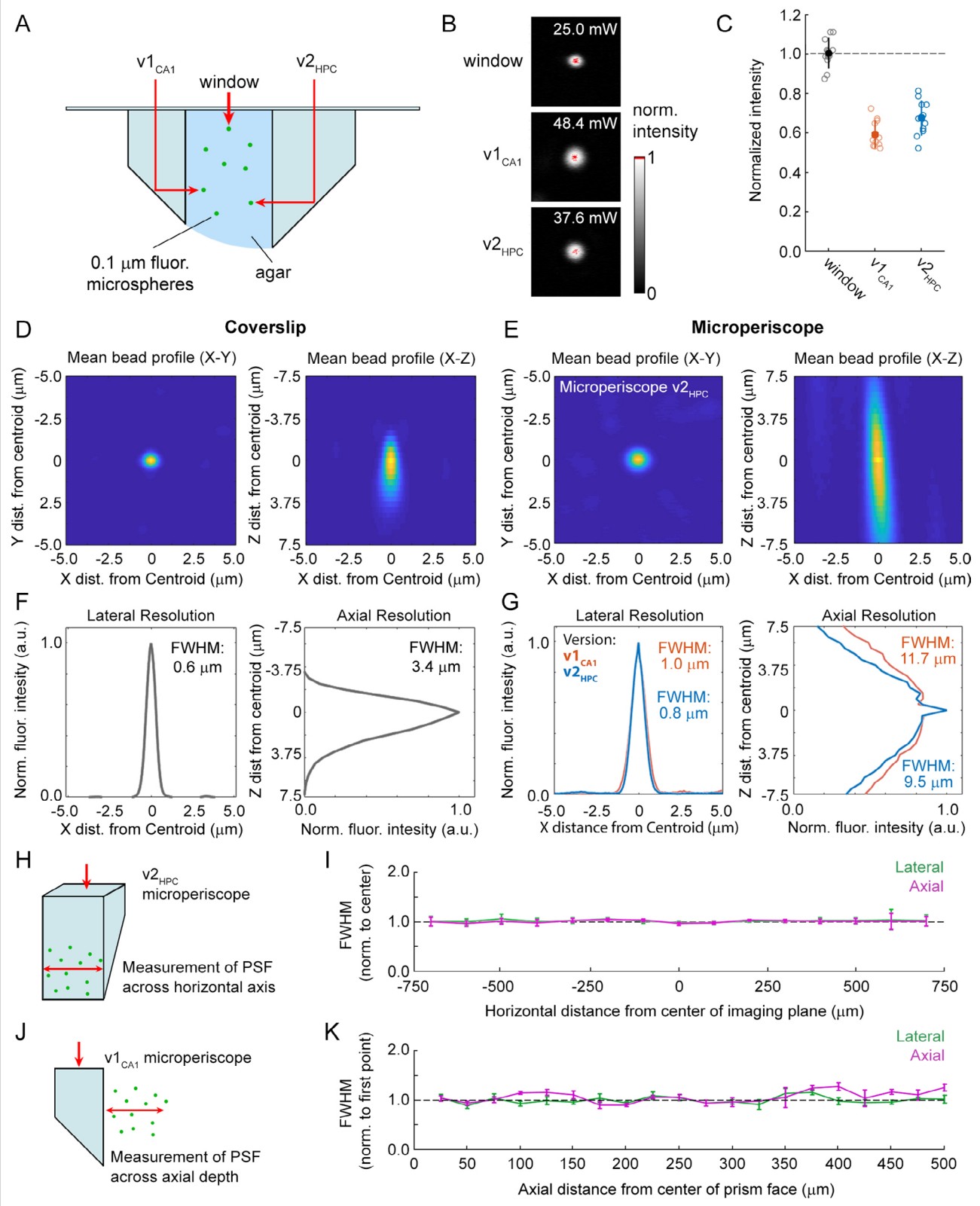

**Figure 2.** Optical characterization of cranial window and microperiscopes. (**A**) Schematic of signal yield and resolution characterization experiments to allow comparison of microperiscopes (v1$_{CA1}$ and v2$_{HPC}$) and standard coverslip window (0.15 mm thickness) using 0.2 m fluorescent microspheres imaged with a 16×/0.8 NA objective. (**B**) Example microspheres imaged through the window, v1$_{CA1}$ microperiscope, and v2$_{HPC}$ microperiscope. Laser power was gradually increased to the minimum level necessary to saturate the centroid of the microsphere with all other imaging parameters held

*Figure 2 continued on next page*

*Figure 2 continued*

constant. (**C**) Distribution of signal intensity values (*n*=12 microspheres for each condition), measured as the reciprocal of the minimum laser intensity for saturation, and normalized to the mean intensity through the window (v1$_{CA1}$: 59.0% ± 6.6%; v2$_{HPC}$: 67.2% ± 8.5%, mean ± s.d.). (**D**) Average *X-Y* profile (left) and *X-Z* profile (right) of fluorescent microspheres (*n*=58 microspheres) imaged through the window. (**E**) Average *X-Y* profile (left) and *X-Z* profile (right) of fluorescent microspheres (*n*=48 microspheres) imaged through the v2HPC microperiscope (2.5 mm path length through glass). (**F**) Plot of normalized fluorescence intensity profile of *X* dimension (lateral resolution; FWHM = 0.6 μm) and *Z* dimension (axial resolution; FWHM = 3.4 m) through the centroid of the microsphere (*n*=58 microspheres) imaged through the window. (**G**) Plot of normalized fluorescence intensity profile of *X* dimension (lateral resolution; v1$_{CA1}$ FWHM = 1.0 m, orange; v2$_{HPC}$ FWHM = 0.8 m, blue) and *Z* dimension (axial resolution; v1$_{CA1}$ FWHM = 11.7 m; v2$_{HPC}$ FWHM = 9.5 m) through the centroid of the microsphere (*n*=28 microspheres for v1$_{CA1}$, *n*=48 microspheres for v2$_{HPC}$). (**H**) Schematic of the experiment used to characterize lateral and axial resolution across the horizontal axis of the microperiscope. Version v2$_{HPC}$ used for larger horizontal range. (**I**) Lateral (green) and axial (magenta) FWHM (mean ± s.e.m.) of average microsphere profile as a function of horizontal distance from the center of the microperiscope. FWHM values normalized to the mean at the center of the imaging axis (–100 to +100 m). There was no effect of horizontal position on lateral FWHM ($F_{(14, 593)}$=0.53, p=0.91) or axial FWHM ($F_{(14, 593)}$=0.91, p=0.55; one-way ANOVA). (**J**) Schematic of the experiment used to measure resolution as a function of distance from the microperiscope face. Version v1$_{CA1}$ used for larger working distance. (**K**) Lateral (green) and axial (magenta) FWHM (mean ± s.e.m.) of average microsphere profile as a function of distance from the face of the microperiscope, with FWHM values normalized to the closest position (25 m). There was no effect of distance of imaging depth on lateral FWHM ($F_{(19, 109)}$=0.53, p=0.26). There was a difference for axial FWHM ($F_{(19, 109)}$=2.51, p=0.002; one-way ANOVA), though no individual position was significantly different from the first position when corrected for multiple comparisons (Tukey-Kramer post hoc test).

neurons (*Figure 1C*) and sub-micron morphological structures (*Figure 3*) with laser power well below the photodamage threshold.

## Imaging spines on the apical dendrites of CA1

Dendritic spines are highly dynamic and motile structures that serve as the postsynaptic sites of excitatory synapses in the hippocampus (*Engert and Bonhoeffer, 1999*; *Harris, 1999*). Previous in vitro studies suggest a role for dendritic spines in structural and functional plasticity, but the transient and dynamic nature of these structures make them ideally suited for being studied in vivo (*Yuste and Bonhoeffer, 2001*; *Yang et al., 2009*; *Attardo et al., 2015*). Although existing techniques allow imaging of spines on the basal CA1 dendrites near the surface of the hippocampus (*Attardo et al., 2015*; *Mizrahi et al., 2004*; *Pfeiffer et al., 2018*), as well as transverse offshoots of apical dendrites (*Gu et al., 2014*; *Ulivi et al., 2019*), imaging hippocampal neurons along the major somatodendritic axis has not previously been possible. Using the microperiscope, we were able to track apical dendritic spine dynamics along the dendrite for CA1-3 neurons in awake mice, though we focused on CA1 dendrites in this study.

In order to visualize apical dendrites in CA1 neurons, we implanted a cohort of Thy1-GFP-M mice, sparsely expressing GFP in a subset of pyramidal neurons (*Feng et al., 2000*), with v1$_{CA1}$ microperiscopes (*Figure 3A*; *n*=7 mice). We focused on CA1 apical dendrites, but both apical and basal dendrites could be imaged in CA1-3 neurons. Although it is theoretically possible to image DG dendritic structures using the microperiscope, the Thy1-GFP-M mouse line has dense expression throughout DG (*Figure 1C and D*), which prevented clear identification of distinct processes. To resolve individual spines along the dendrite, high-resolution images were taken from several axial planes spanning the segment, and a composite image was generated using a weighted average of individual planes (*Figure 3B*; see Materials and methods). We reduced noise by filtering and binarizing, and isolated dendrites of interest for tracking across days (*Figure 3C and D*; *Figure 3—figure supplement 1A*; see Materials and methods).

Previous studies have shown that dendritic spines fall into four major morphological subtypes: filopodium, thin, mushroom, and stubby (*Chang and Greenough, 1984*; *Rodriguez et al., 2008*; *Son et al., 2011*). We found that our resolution was sufficient to classify dendritic spines into their relative subtypes and evaluate density and turnover based on these parameters (*Figure 3E and F*; *Figure 3—figure supplement 1B*). Consistent with previous studies, we found a non-uniform distribution of dendritic spines: 30.4% thin, 41.0% stubby, 26.2% mushroom, and 2.3% filopodium ($F_{(3,100)}$ = 51.47, p<0.0001, one-way ANOVA; *Figure 3E*). The low proportion of filopodium found in this and previous 2P imaging and histological studies (*Lendvai et al., 2000*; *Risher et al., 2014*) (2–3%), as compared to electron microscopy studies (*Fiala et al., 1998*; *Stewart et al., 2005*) (~7%), may result from the narrow width of these structures causing them to fall below the detection threshold (*Attardo et al., 2015*; *Pfeiffer et al., 2018*; *Gu et al., 2014*; *Castello-Waldow et al.,*

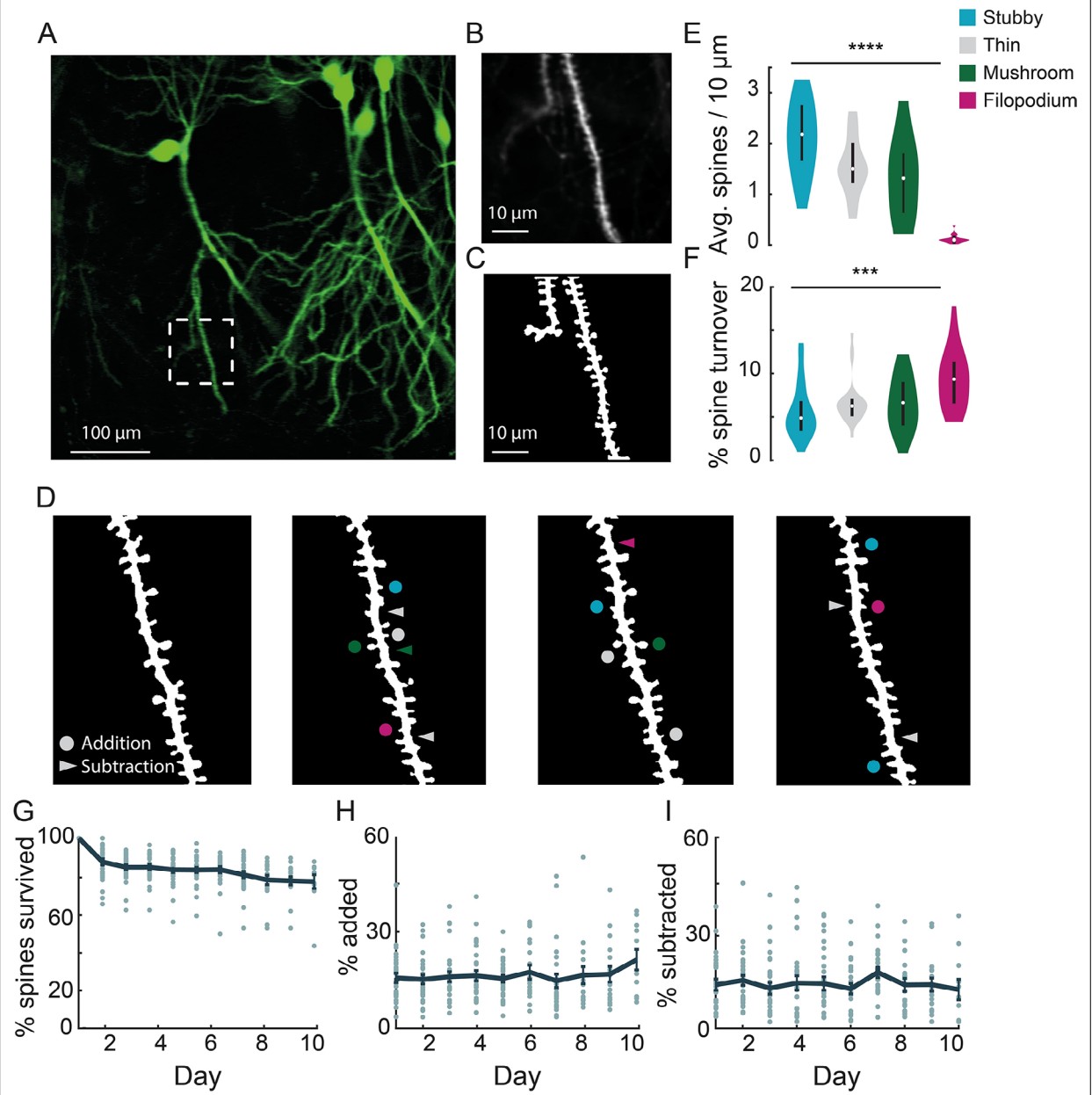

**Figure 3.** Chronic imaging of spine morphology in CA1 apical dendrites. (**A**) Average projection of CA1 neurons sparsely expressing a GFP reporter (Thy1-GFP-M) imaged through the v1$_{CA1}$ microperiscope. Scale bar = 100 µm. (**B**) Weighted projection (see Materials and methods) of the apical dendrites shown in the dashed box of (**A**). Scale bar = 10 µm. (**C**) Filtered and binarized image (*Figure 3—figure supplement 1A*; see Materials and methods) of the dendrites in (**B**) to allow for the identification and classification of individual dendritic spines. Scale bar = 10 µm. (**D**) Tracking CA1 dendritic spines over consecutive days on a single apical dendrite. Arrowheads indicate subtracted spines and circles indicate added spines. Colors indicate spine type of added and subtracted spines: filopodium (magenta), thin (gray), stubby (blue), and mushroom (green). (**E**) Average number of spines per 10 µm section of dendrite, for each of the four classes of spine (n=26 dendrites from 7 mice); one-way ANOVA, $F(3,100)$ = 51.47, ****p<0.0001. Error bars indicate the interquartile range (75th percentile minus 25th percentile) and circle is median. (**F**) Percent spine turnover across days in each spine type (n=26 dendrites from 7 mice); one-way ANOVA, $F(3,100)$ = 7.17, ***p<0.001. Error bars indicate the interquartile range (75th percentile minus 25th percentile) and circle is median. (**G**) Spine survival fraction across processes (n=26 dendrites from 7 mice) recorded over 10 consecutive days. (**H**) Percent of spines added between days over 10 days of consecutive imaging. (**I**) Percent of spines subtracted between days over 10 days of consecutive imaging.

The online version of this article includes the following figure supplement(s) for figure 3:

**Figure supplement 1.** Dendritic morphology image processing pipeline and spine classification.

**Figure supplement 2.** Long-term imaging of the same dendrite.

*2020*). Consistent with previous work (*Grutzendler et al., 2002*; *Trachtenberg et al., 2002*; *Holt-maat et al., 2005*), we found that particular classes, such as filopodium, had a high turnover rate, while other classes were more stable across sessions ($F$(3,100) = 7.17, p<0.001, one-way ANOVA; *Figure 3F*).

We found that total turnover dynamics reflect 15.0% ± 2.0% spine addition and 13.0% ± 1.9% spine subtraction across consecutive days, representing a high degree of instability (*Figure 3H, I*). We computed the survival fraction, a measure indicating the fraction of spines still present from day 1 (*Yang et al., 2009*; *Attardo et al., 2015*; *Pfeiffer et al., 2018*). Although daily spine addition and subtraction was approximately 15.0% (*Figure 3H, I*), the survival fraction curve yields a more conservative estimate of turnover dynamics (*Figure 3G*). Across 10 days, we found a 23.5% net loss in original spines, indicating that most spine turnover takes place within an isolated population of transient spines. Both our cumulative turnover and survival fraction results were generally similar to previous findings from basal dendrites in CA1 (*Attardo et al., 2015*; *Mizrahi et al., 2004*; *Pfeiffer et al., 2018*), indicating that apical and basal dendrites exhibit similar spine dynamics. It should be noted that some previous studies used super-resolution techniques to detect smaller spines and reduce optical merging. Thus, our results may suggest an inflated degree of stability due to resolution limitations that prevent capture of filipodia and other small spine structures (*Attardo et al., 2015*; *Pfeiffer et al., 2018*; *Gu et al., 2014*; *Castello-Waldow et al., 2020*). Despite these limitations, we found that we could identify the same dendritic processes over long time periods (up to 150 days; *Figure 3—figure supplement 2*), allowing for longitudinal experiments tracking isolated dendritic structures over long time intervals.

## Recording PCs and SCs in CA1, CA3, and DG

Much of the experimental work testing the hypothesized roles of CA1, CA3, and DG neurons has come from PC recordings (*Leutgeb et al., 2004*; *Leutgeb et al., 2005*; *Leutgeb et al., 2007*; *Neunuebel and Knierim, 2014*; *van Dijk and Fenton, 2018*; *Stefanini et al., 2020*). While the results of these studies have been instrumental, it has not been possible to measure activity throughout the transverse hippocampal circuit in the same animals. We therefore investigated the ability of our microperiscope to record from PCs in each of the hippocampal subfields during exploration of a spatial environment.

To measure functional responses, we implanted $v2_{HPC}$ microperiscopes in transgenic mice expressing GCaMP6s in glutamatergic neurons (*Chen et al., 2013*; *Daigle et al., 2018*) (see Materials and methods). As with the Thy1-GFP-M mice, we were able to image neurons from CA1-CA3, and DG in the same animal (*Figure 4A and B*; *Video 1*). In some cases, depending on microperiscope placement, we were able to record from all three areas simultaneously (*Figure 4—figure supplement 1*). In all HPC subfields, we found normal calcium dynamics with clear transients (*Figure 4C, D*; *Video 2* and *Video 3*). The imaging fields remained stable, and the same field could be imaged over 100 days later (*Figure 4E*). In addition, microperiscope implantation allowed measurement of neural responses from mossy cells in the DG (*Figure 4—figure supplement 1* and *Figure 4—figure supplement 2*) and, depending on microperiscope placement, simultaneous imaging of deep-layer cortical neurons in parietal cortex (*Figure 4—figure supplement 2B*).

As 2P microscopy generally requires the animal to be head-fixed, the behavioral assays used to probe PC activity are limited. Previous work has made use of virtual reality (VR) (*Hainmueller and Bartos, 2018*; *Dombeck et al., 2010*), however it remains unclear how similar rodent hippocampal activity in real-world environments is to that in VR (*Aronov and Tank, 2014*; *Aghajan et al., 2015*). To study PC activity in a physical environment, our head-fixed mice explored a carbon fiber arena that was floated using an air table (*Kislin et al., 2014*) (*Figure 5A*; see Materials and methods). The mice were thus able to navigate the physical chamber by controlling their movement relative to the floor. Although this approach lacks the vestibular information present in real-world navigation, it captures somatosensory and proprioceptive information missing from virtual environments (*Go et al., 2021*; *Aghajan et al., 2015*; *Ravassard et al., 2013*). Moreover, recent work has found that place field width and single-cell spatial information using this approach is comparable to the responses of free foraging animals (*Go et al., 2021*). For measurement of place coding, we allowed mice to navigate a curvilinear track over the course of 20–40 min (*Video 4*). As found previously (*Go et al., 2021*), using a curvilinear track allowed for robust sampling of the spatial environment and improved place field localization, though they could also be measured in the open field. In order to measure spatial properties of the

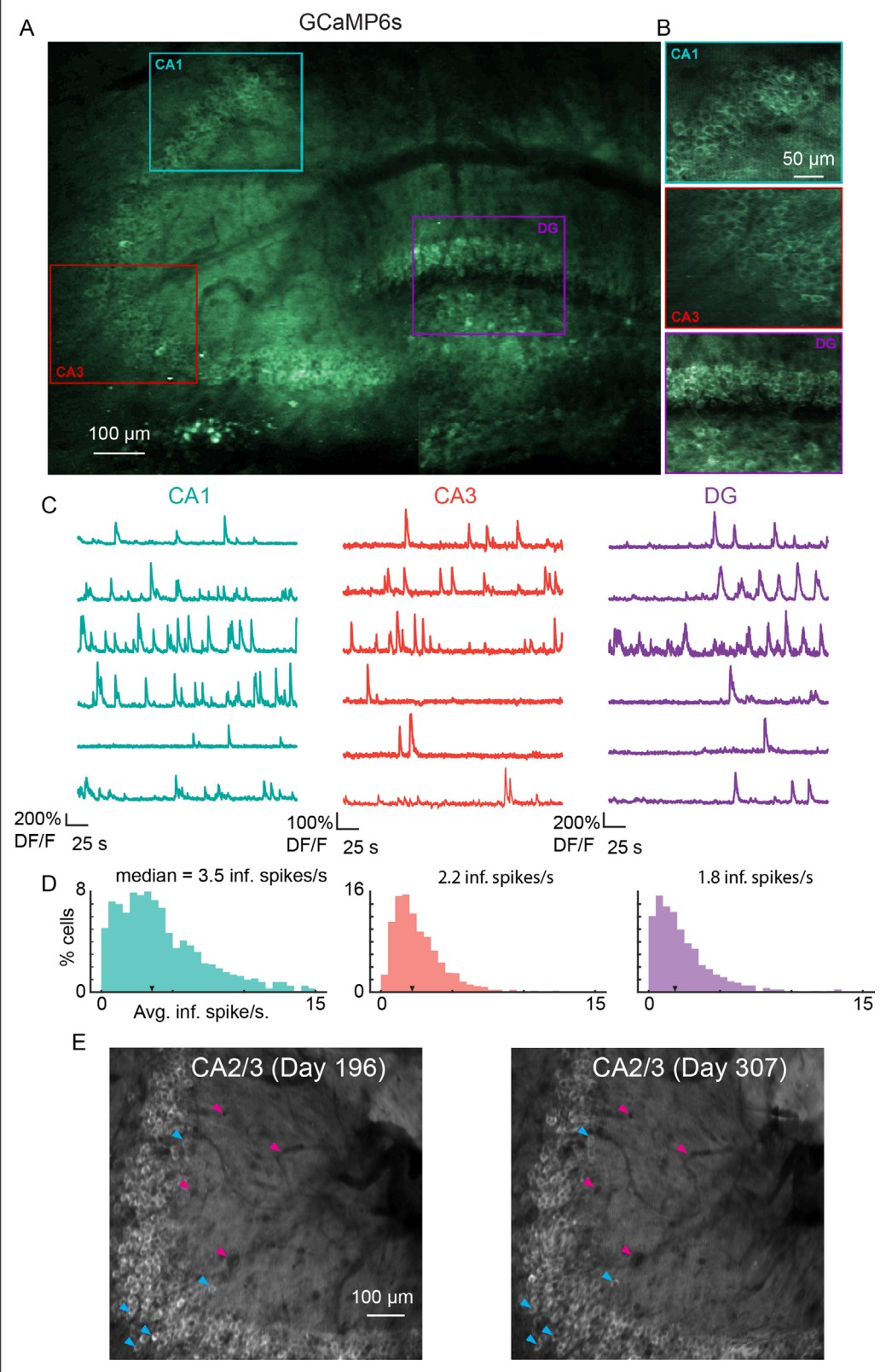

**Figure 4.** Microperiscope imaging of calcium dynamics across subfields in awake mice. (**A**) Example tiled image (as in *Figure 1C*) for a single panexcitatory transgenic GCaMP6s mouse (Slc17a7-GCaMP6s). Scale bar = 100 µm. (**B**) Enlarged images of each hippocampal subfield (CA1, CA3, DG) corresponding to the rectangles in (**A**). Scale bar = 50 µm. (**C**) Example GCaMP6s normalized fluorescence time courses (% *DF/F*) for identified cells in each

*Figure 4 continued on next page*

*Figure 4 continued*

subfield. (**D**) Distribution of average inferred spiking rate during running for each hippocampal subfield (CA1, CA3, DG; see Materials and methods). Median is marked by black arrowhead. (**E**) Example average projection of CA2/3 imaging plane 196 days post implantation (left) and 111 days later. Image was aligned using non-rigid registration (see Materials and methods) to account for small tissue movements. Magenta arrowheads mark example vasculature and blue arrowheads mark example neurons that are visible in both images. Scale bar = 100 μm.

The online version of this article includes the following figure supplement(s) for figure 4:

**Figure supplement 1.** Simultaneous imaging of all three hippocampal subfields.

**Figure supplement 2.** Additional applications for microperiscope imaging.

hippocampal neurons independent of reward, we relied on exploration rather than active reward administration for sampling of the environment.

To characterize place fields, we recorded from neurons in CA1, CA3, and DG (CA1: *n*=1001; CA3: *n*=832; and DG: *n*=463) in transgenic mice with panexcitatory expression of GCaMP6s (*Figure 5B*; *n*=8 mice). We found PCs by assessing lap-by-lap reliability and goodness-of-fit to a Gaussian in all three subfields (*Figure 5C*; false positive rate in shuffled data = 0%; see Materials and methods). The distribution was in general agreement with previous 2P imaging experiments in mice (*Dong et al., 2021*; *Hainmueller and Bartos, 2018*) (*Figure 5D and G*; CA1: 26.0%; CA3: 18.9%; DG: 14.0%), with fields that spanned the entirety of the track (*Figure 5F*). We found that place field widths were comparable across the three regions (*Figure 5H*; CA1: mean (median) ± s.e.m.=18.8 (19.4) ±0.4 cm; CA3: 18.6 (18.8) ± 0.5 cm; DG: 18.9 (19.5) ±0.8 cm) and that spatial information was highest in CA1, followed by CA3, then DG (*Figure 5I*; CA1: mean (median) ± s.e.m.=0.95 (0.87) ± 0.03 bits/inferred spike; CA3: 0.89 (0.68) ± 0.05 bits/inferred spike; DG: 0.79 (0.61) ± 0.08 bits/inferred spike). The place field width and spatial information of the CA1 PCs were similar to those found in a previous study using the same floating chamber design with a traditional hippocampal window implant (*Go et al., 2021*).

To determine if neurons close to the microperiscope face showed unusual response properties, we imaged functional responses as a function of distance. PC width did not vary significantly as a function of depth of the imaging plane from the face of the microperiscope (*Figure 5—figure supplement 1A*; $F(4, 241)=1.6$, p=0.17, one-way ANOVA). Spatial information did vary as a function of depth ($F(4, 1872)=15.9$, p=$8.6 \times 10^{-13}$, one-way ANOVA), but there was no systematic effect (*Figure 5—figure supplement 1B*). Decay constant for the fitted transients (*Friedrich et al., 2017*) did have a significant relationship with imaging depth ($F(4, 1872)=29.8$, p=$4.5 \times 10^{-24}$), which may be due to aberrant activity in a small subset of neurons

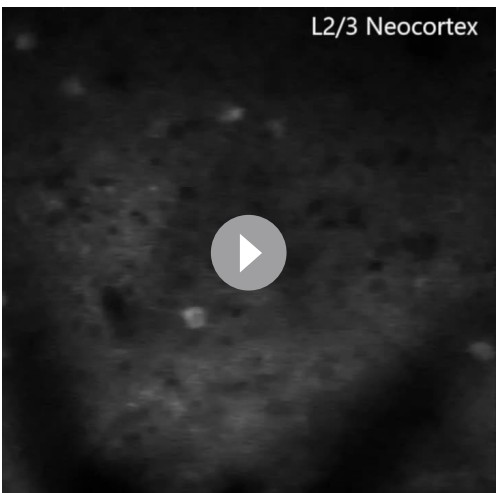

**Video 1.** Demonstration of two-photon imaging of an Slc17a7-GCaMP6s mouse through the microperiscope. Recording starts in the superficial cortex in front of microperiscope, then moves through the microperiscope to the hippocampus, zooming in on subfields CA1, CA3, and DG.
https://elifesciences.org/articles/75391/figures#video1

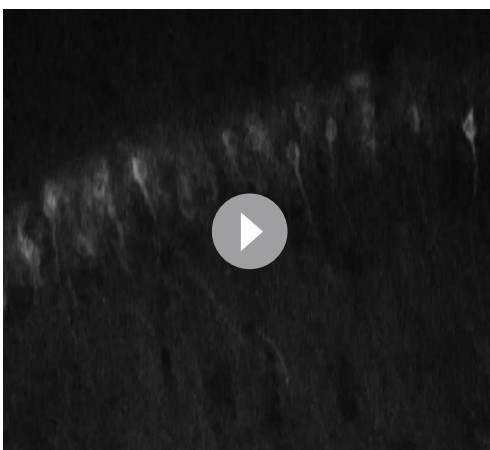

**Video 2.** Calcium activity in subfield CA1 in an Slc17a7-GCaMP6s mouse imaged through the microperiscope.
https://elifesciences.org/articles/75391/figures#video2

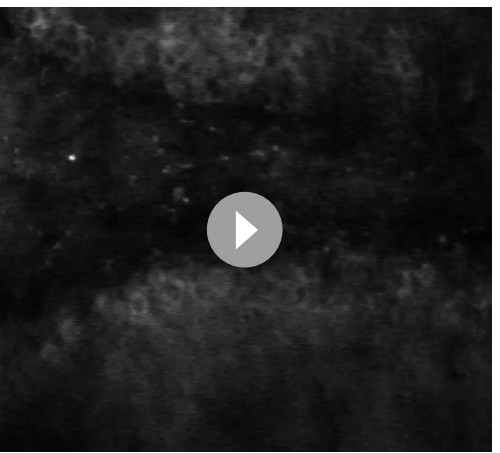

**Video 3.** Calcium activity in subfield DG in an Slc17a7-GCaMP6s mouse imaged through the microperiscope. https://elifesciences.org/articles/75391/figures#video3

close ( ≤ 130 μm) to the microperiscope face (*Figure 5—figure supplement 1C*).

As cells responsive to speed have recently been found in the medial EC (*Kropff et al., 2015*; *Iwase et al., 2020*) and CA1 (*Kropff et al., 2015*; *Iwase et al., 2020*), we also identified neurons as SCs if their activity was significantly related to running speed (*Kropff et al., 2015*) (see Materials and methods; average speed per recording: CA1, 71.3 ± 33.7 mm/s; CA3, 76.2 ± 32.1; DG, 101.1 ± 49.0; Two-sample Kolmogorov-Smirnov test: CA1-CA3, p=0.90; CA3-DG, p=0.53; CA1-DG, p=0.32). We found SCs in CA1, CA3, and DG, with all areas having cells that showed both increased and decreased activity with higher running speeds (*Figure 5E*). SCs were most abundant in DG (*Figure 5G*; CA1: 9.4%; CA3: 13.5%; DG: 30.2%), consistent with recent work that found it was possible to decode the speed of freely moving animals from the activity of DG, but not CA1 (*Stefanini et al., 2020*).

## Distribution of PC properties along the DG-to-CA1 axis

Recent work has suggested that PC properties are heterogeneously distributed along the extent of the DG-to-CA1 axis. In particular, place field width and spatial information have been found to vary among different subregions of CA3, with distal CA3 (dCA3) and CA2 having lower spatial information and larger place fields than medial CA3 (mCA3) (*Lee et al., 2015*; *Lu et al., 2015*; *Mankin et al., 2015*), and proximal CA3 (pCA3) having values most similar to DG (*Neunuebel and Knierim, 2014*). Such distributions could be supported by known anatomical gradients in connectivity of CA3 (*Claiborne et al., 1986*; *Ishizuka et al., 1990*; *Li et al., 1994*; *Ishizuka et al., 1995*; *Witter, 2007*). However, given that these studies required separate animals for the recording of each location along the DG-to-CA1 axis, and that electrophysiology has limited spatial resolution along the transverse axis, we used our microperiscopes to measure these properties throughout the DG-to-CA1 axis in the same mice with high spatial resolution.

Using the v2$_{HPC}$ microperiscope, we simultaneously imaged from several hundred cells (mean ± s.d.; 215 ± 74 neurons per recording; total: 1075 neurons) extending from pCA3 to pCA1 (*Figure 6A*). Recordings from distinct imaging planes in different mice (*n*=5) were compared by calculating the distance of individual cells from the inflection point of the DG-to-CA1 transverse axis (*Figure 6A*; see Materials and methods). In agreement with previous studies (*Lee et al., 2015*; *Lu et al., 2015*; *Mankin et al., 2015*), we found a non-uniform distribution of spatial information along this axis (*Figure 6B and C*; $F(4, 1047)=5.10$, p=$4.6 \times 10^{-4}$; general linear F-test against a flat distribution with the same mean). In particular, we found that pCA3 cells had spatial information that was closer in value to those in DG than mCA3 (*Figure 6B and C*; *Figure 5I*), and that mCA3 had spatial information that was greater than dCA3/CA2 (*Figure 6B and C*). We found that place field widths were narrowest in mCA3, although we failed to find a statistically significant non-uniform distribution with respect to place field width across the extent of the DG-to-CA1 axis (*Figure 6—figure supplement 1*; $F(4, 206)=1.63$, p=0.17, general linear F-test against a flat distribution with the same mean).

## Discussion

The microperiscope hippocampal imaging procedure we developed allows researchers, for the first time, to chronically image neuronal structure and functional activity throughout the transverse hippocampal circuit in awake, behaving mice. This approach builds on microprism procedures developed for imaging cortical columns (*Chia and Levene, 2009*; *Andermann et al., 2013*; *Low et al., 2014*), allowing multiple hippocampal subfields in the same animal to be accessed optically. Using the

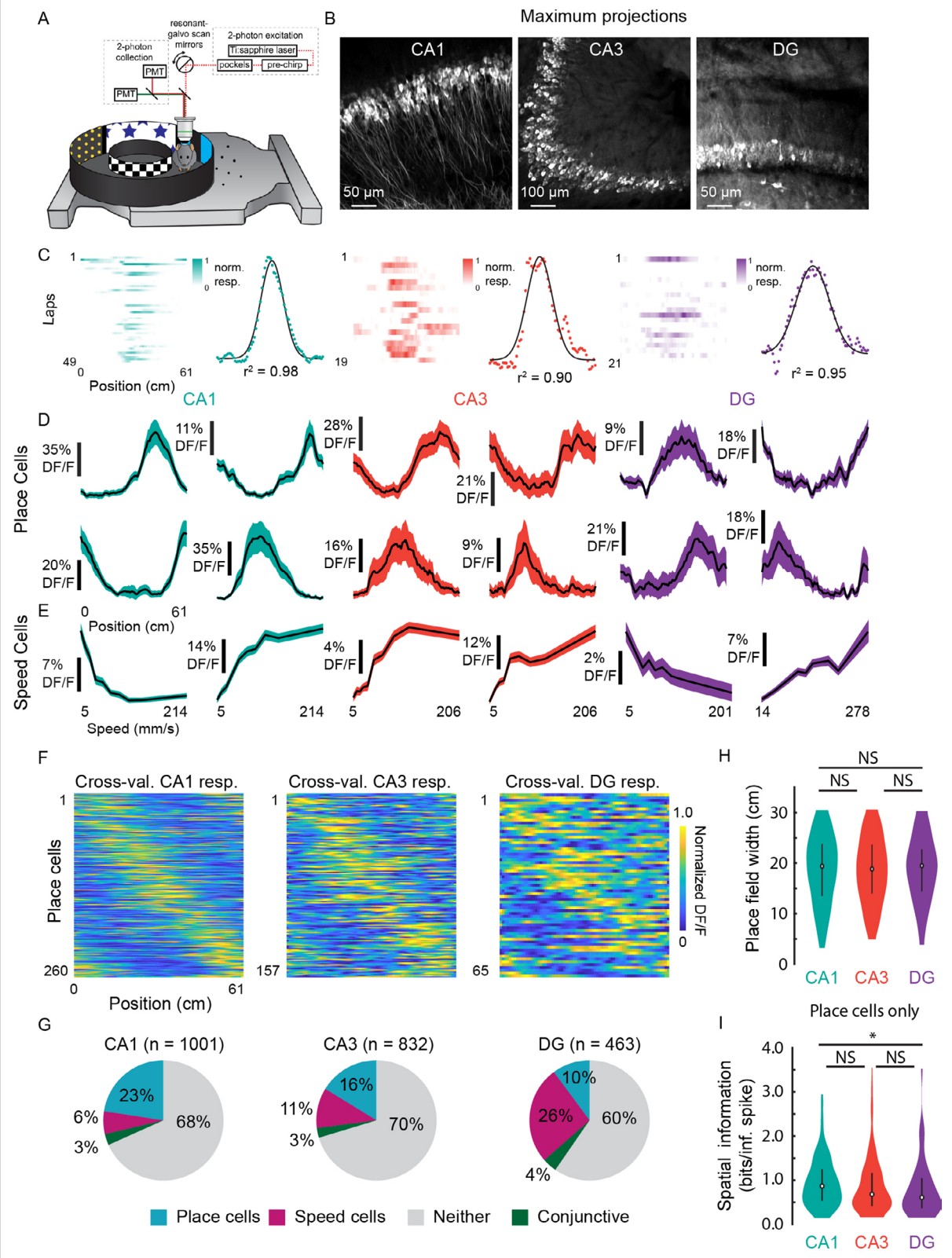

**Figure 5.** Prevalence of place (PCs) and speed cells (SCs) across hippocampal subfields. (**A**) Schematic of air-lifted carbon fiber circular track (250 mm outer diameter) that the mice explored during imaging (***Video 4***). Four sections of matched visual cues lined the inner and outer walls. (**B**) Example maximum projections of GCaMP6s-expressing neurons in each subfield. Scale bar = 50, 100, 50 µm, respectively. (**C**) Lap-by-lap activity and Gaussian fits, with $r^2$ value reported, of example PCs for each hippocampal subregion to illustrate our process of identifying PCs (see Materials and methods).

*Figure 5 continued on next page*

*Figure 5 continued*

(**D**) Examples of mean normalized calcium response (% *DF/F*) vs. position along the circular track for four identified PCs in each subregion. Shaded area is s.e.m. (**E**) Examples of mean calcium response (% *DF/F*) vs. speed along the circular track for two identified SCs in each subregion. Shaded area is s.e.m. (**F**) Cross-validated average responses (normalized % *DF/F*) of all PCs found in each subfield, sorted by the location of their maximum activity. Responses are plotted for even trials based on peak position determined on odd trials to avoid spurious alignment. (**G**) Distribution of cells that were identified as PCs, SCs, conjunctive PC + SCs, and non-coding in each subregion. (**H**) Distribution of place field width for all PCs in each subregion. Error bars indicate the interquartile range (75th percentile minus 25th percentile) and circle is median. Two-sample Kolmogorov-Smirnov test: CA1-CA3, p=0.67; CA3-DG, p=0.70; CA1-DG, p=0.51. NS, not significant (p>0.05). (**I**) Distribution of spatial information (bits per inferred spike) for all PCs in each subregion. Error bars and circle are the same as in (**H**). Two-sample Kolmogorov-Smirnov test: CA1-CA3, p=0.05; CA3-DG, p=0.36; CA1-DG, p=0.01. *p<0.05.

The online version of this article includes the following figure supplement(s) for figure 5:

**Figure supplement 1.** Place cell properties as a function of distance from the face of the microperiscope.

microperiscope, we were able to resolve spines on the apical dendrites of CA1 pyramidal cells and track them across time. Additionally, we were able to characterize PCs and SCs in all three hippocampal subfields and investigate their anatomical distribution across subfields.

## Comparison to other methods

Over the past five decades, electrophysiology has been the principal tool used to study the hippocampus. Electrophysiological recordings have much higher temporal resolution than calcium imaging and can directly measure spikes, but have limited spatial resolution. Our approach allows large scale imaging of neurons across multiple hippocampal subfields, with known spatial and morphological relationships. In addition, our approach allows genetically controlled labeling of particular cell types, imaging of cellular structures such as dendrites and spines, and unequivocal tracking of functional and structural properties of the same cells across time, none of which are possible with existing electrophysiological approaches.

Several approaches making use of optical imaging have been developed for use in the hippocampus. These include gradient index lenses (*Levene et al., 2004*; *Barretto et al., 2011*), and cannulas that can be combined with both one-photon head-mounted microendoscopes (*Ghosh et al., 2011*; *Ziv et al., 2013*; *Cai et al., 2016*) and 2P imaging (*Hainmueller and Bartos, 2018*; *Dombeck et al., 2010*; *Sheintuch et al., 2017*). These methods have been limited to horizontal imaging planes, making it difficult to image CA3 and DG, and intractable to image all three subfields in the same animal. While all of these methods cause damage to the brain, and some require the aspiration of the overlying cortex, the damage is largely restricted to superficial hippocampus. This contrasts with the implantation of our microperiscope, which is inserted into the septal end of the hippocampus and necessarily causes some damage to the structure. Despite this, we find normal response properties (*Figure 4*, *Figure 5—figure supplement 1*), including selectivity for location and speed, in CA1-CA3 and DG (*Figure 5*). Damage in the direction orthogonal to the transverse axis caused by the implantation of the microperiscope is similar to that caused by microprisms in cortex (*Andermann et al., 2013*), as glial markers were found to decay to baseline levels with increased distance from the face of the microperiscope (*Figure 1E and F*). We showed that, following successful implantation, the imaging fields were stable, and the same cells could be imaged up to 3 months later (*Figure 4E*). However, we emphasize that tissue damage caused by the microperiscope assembly to the hippocampus should be taken into consideration when planning experiments and interpreting results, and that imaging planes should be at least 150 µm away from microperiscope face to avoid aberrant responses.

We found that imaging through the microperiscope had some effect on signal intensity and axial resolution. The effect depended not only on the path length through the glass, but also on the

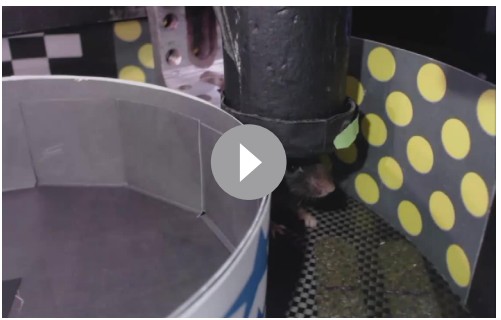

**Video 4.** Mouse running in the floating chamber circular track. Ambient light is higher than usual levels for improved video quality.

https://elifesciences.org/articles/75391/figures#video4

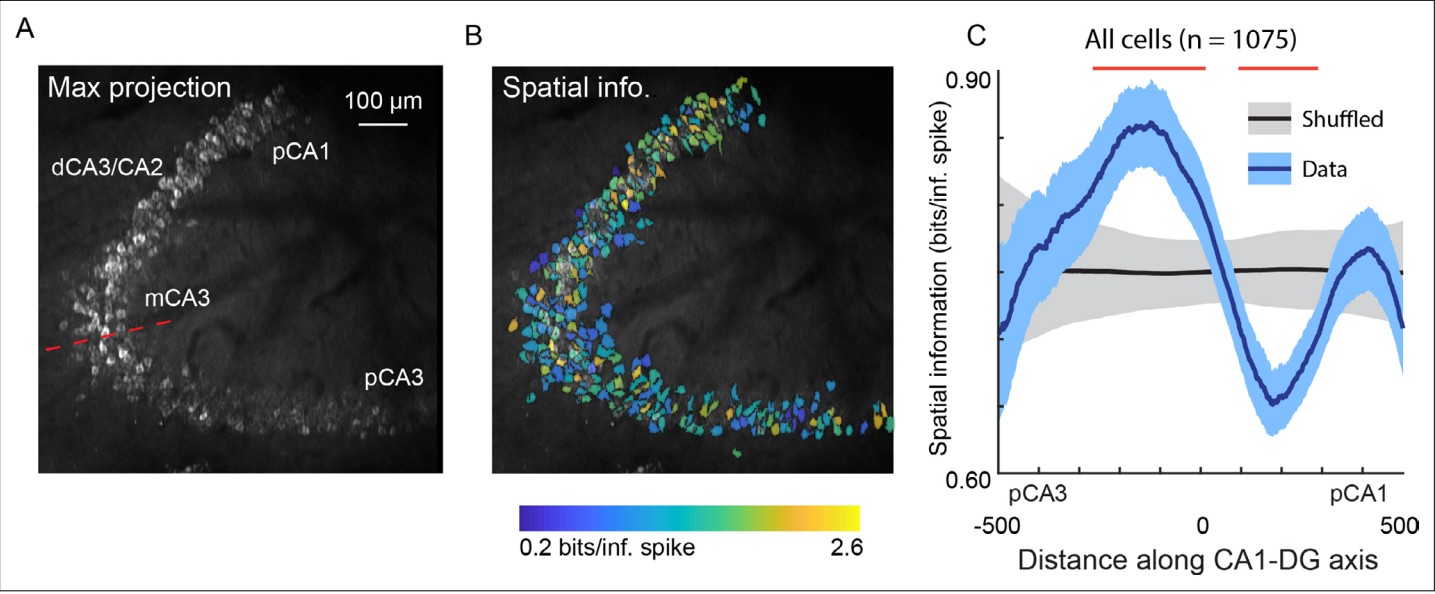

**Figure 6.** Spatial information of neurons varies along the DG-to-CA1 axis. (**A**) Maximum projection of an example DG-to-CA1 axis recording. Approximate locations of CA3 and CA1 subfields are labeled. Inflection point labeled with red line. Scale bar = 100 µm. (**B**) Spatial information (bits/inferred spike), pseudo-colored on a logarithmic scale, for each neuron, overlaid on the maximum projection in (**A**). (**C**) Spatial information, as a function of distance along the DG-to-CA1 axis (pCA3 to dCA1), calculated with a sliding window; mean ± bootstrapped s.e.m.; real data (blue), shuffled control (black). Red lines indicate values that are outside the shuffled distribution (p<0.05). A general linear F-test against a flat distribution with the same mean revealed significant non-uniformity: $F(4, 1047)=5.10$, $p=4.6 \times 10^{-4}$ (cells with distance greater than 600 or less than –600 were not included in the statistical analysis).

The online version of this article includes the following figure supplement(s) for figure 6:

**Figure supplement 1.** Place field width does not significantly vary along the DG-to-CA1 axis.

geometry of the microperiscope. For example, the larger $v2_{HPC}$ microperiscope had slightly better optical properties than the smaller $v1_{CA1}$ microperiscope (*Figure 2*), likely due to reduced beam clipping. The optical properties should be considered along with the potential tissue damage during the planning of experiments. Moreover, further customizing the microperiscope geometry for particular applications may help optimize signal quality and minimize damage to neural structures.

To assess the ability of our optical and behavioral experimental methods to capture spatial coding, we chose to measure place fields in the absence of reward, which is known to modulate hippocampal activity (*Gauthier and Tank, 2018*). Recent work has found that PCs are significantly less stable in the absence of reward (*Krishnan et al., 2020*; *Pettit et al., 2022*), which may explain the lower lap-wise reliability in our experiments compared to similar experiments using the same behavioral apparatus (*Go et al., 2021*). However, we found similar percentages of place fields as other optical imaging studies, despite using a strict criteria for PC inclusion (see Materials and methods), and these properties did not change systematically as a function of distance from the face of the periscope (*Figure 5—figure supplement 1A, B*). This provides further support that the functional properties of the hippocampal circuitry remained intact when using the microperiscope. However, we note that it is possible that the PC characteristics we measured might be altered in the presence of reward, and that this decision should be taken into account when interpreting the results.

## Structural and functional properties along the transverse hippocampal circuit

We utilized the microperiscope in two experiments that would not have been possible with existing methods: (1) we tracked the spines along apical CA1 dendrites in vivo (*Figure 3*); (2) we simultaneously recorded from PCs along the extent of the DG-to-CA1 transverse axis (*Figure 6*).

Several studies have tracked the spines of basal CA1 dendrites in vivo by imaging the dorsal surface of the hippocampus (*Attardo et al., 2015*; *Mizrahi et al., 2004*; *Pfeiffer et al., 2018*; *Ulivi et al., 2019*; *Castello-Waldow et al., 2020*). However, imaging spines along the major dendritic

axis has not been accomplished in vivo. Given that these spines make up the majority of the input to CA1 pyramidal neurons, there is a significant need for understanding their dynamics, and how the dynamics differ as a function of position along the somatodendritic axis. Using the microperiscope, we tracked isolated apical dendrites for up to 10 consecutive days (*Figure 3D*). We found considerable addition and subtraction across days, indicating dynamic turnover in apical dendrites (*Figure 3H, I*). However, consistent with previous studies in basal spines (*Attardo et al., 2015*; *Mizrahi et al., 2004*; *Pfeiffer et al., 2018*), we found the majority of spines (76.5%) survived throughout the imaging period (*Figure 3G*). This high survival fraction suggests that the daily turnover rates we observe are reflective of a distinct pool of unstable transient spines, while the majority of spines are moderately stable, and remain present over a longer timescale. It should be noted that some prior results were obtained using super-resolution microscopy techniques that are more suited to capturing the dynamics of filipodia and small spines, and thus our results may reflect an inflated degree of stability (*Attardo et al., 2015*; *Pfeiffer et al., 2018*; *Ulivi et al., 2019*). The microperiscope is in principle compatible with super-resolution microscopy, and future imaging studies that combine the two could characterize apical dendritic spine dynamics with improved accuracy (*Attardo et al., 2015*; *Pfeiffer et al., 2018*; *Ulivi et al., 2019*). Taken together, our results add to the increasingly appreciated idea that, even in the absence of salient learning and reward signals, dendritic spines are dynamic and unstable structures (*Attardo et al., 2015*; *Chambers and Rumpel, 2017*; *Mongillo et al., 2017*). Understanding the nature and timescales of these dynamics has significant implications for the reported instability of PCs (*Mankin et al., 2012*; *Hainmueller and Bartos, 2018*; *Ziv et al., 2013*; *Kentros et al., 2004*; *Kinsky et al., 2018*; *Mau et al., 2020*).

Previous studies have found gradients of connectivity in CA3 (*Claiborne et al., 1986*; *Ishizuka et al., 1990*; *Li et al., 1994*; *Ishizuka et al., 1995*; *Witter, 2007*), suggesting that there may be functional gradients as well. Electrophysiological recordings have indeed found that spatial information and place field width vary as a function of distance along the DG-to-CA1 axis (*Lee et al., 2015*; *Lu et al., 2015*; *Mankin et al., 2015*), and that the most proximal part of CA3 is functionally more similar to DG than to the rest of CA3 (*Neunuebel and Knierim, 2014*). This has led to a more nuanced understanding of the hippocampal circuit, with coarse anatomical subdivisions having finer functional subdivisions. However, given that these previous studies relied on recordings in different animals for each location along the DG-to-CA1 axis, and given that electrodes have poor spatial resolution in the transverse axis, the results must be cautiously interpreted. Using the microperiscope, we imaged several hundred neurons in multiple mice along the extent of the DG-to-CA1 axis (*Figure 6A*). The location of each neuron relative to the CA3 inflection point could be easily identified, allowing for unequivocal characterization of spatial information and place field width along the DG-to-CA1 axis across animals. Similar to the previous studies (*Lee et al., 2015*; *Lu et al., 2015*; *Mankin et al., 2015*), we found a non-uniform distribution of spatial information (*Figure 6B and C*). Place field width also appeared to vary along the DG-to-CA1 axis, though we failed to find statistically significant non-uniformity with respect to place field width (*Figure 6—figure supplement 1B*). Indeed, we found that spatial information, but not place field width, differed significantly between CA1, CA3, and DG (*Figure 5H, I*). Given the high spatial resolution and ability to simultaneously record from cells across the DG-to-CA1 axis using this approach, our results strengthen the hypothesis of distributed spatial coding across, and within, hippocampal subfields.

## Future applications

This paper explored a few possible uses for the microperiscope in interrogating the hippocampal circuit. However, there are a number of candidate applications we did not pursue that could reveal novel insight into hippocampal function. Here, we describe several of these, chosen to highlight the utility of the microperiscope in addressing questions that are challenging or intractable with existing approaches: (1) The microperiscope allows for the recording from multiple hippocampal subfields simultaneously (*Figure 6*; *Figure 4—figure supplement 1*), allowing investigation of interactions between neurons in different subfields during behavior. This could also be useful for determining the effect of neuron- or subfield-specific optogenetic manipulations on downstream subfields. (2) The microperiscope enables the investigation of local circuits by allowing morphological and genetic identification of different cell types. This includes identifying genetically distinct hippocampal neurons (e.g. CA2 neurons, specific interneuron subtypes), as well was identifying particular cell types by position

or morphological characteristics (e.g. mossy cells in DG; *Figure 4—figure supplement 2A*). As these distinct cell types play important roles in hippocampal function (*Hitti and Siegelbaum, 2014*; *Wilson and McNaughton, 1993*; *Danielson et al., 2017*; *GoodSmith et al., 2017*; *Senzai and Buzsáki, 2017*; *Mongillo et al., 2018*), having access to them will enable a greater understanding of the hippocampal circuit. (3) The microperiscope can be combined with retrograde-transported viruses to allow projection-based cell identification, making it possible to identify neurons that project to specific downstream targets. Since hippocampal neurons that project to different brain regions have been found to exhibit distinct functional properties (e.g. neurons in ventral CA1 that project to the nucleus accumbens shell have been implicated in social memory; *Okuyama et al., 2016*), the union of these tools will be a powerful means for understanding hippocampal outputs. (4) Finally, the microperiscope provides access to the entire dendritic tree of pyramidal neurons (*Figures 1C, 3A and 5B*; *Video 2*), giving optical access to dendritic signaling over a much larger spatial extent than has been previously possible (*Sheffield and Dombeck, 2015*; *Sheffield et al., 2017*; *Adoff et al., 2021*). By sparsely expressing calcium or glutamate sensors in hippocampal pyramidal neurons, spines throughout the somatodendritic axis could be imaged, allowing determination of how place field responses arise from the responses of individual spines, similar to experiments in visual cortex investigating the cellular origin of orientation tuning from synaptic inputs (*Jia et al., 2010*; *Wilson et al., 2016*).

Combined with existing electrophysiological and imaging approaches, imaging of the transverse hippocampal circuit with microperiscopes will be a powerful tool for investigating hippocampal circuitry, structural dynamics, and function.

# Materials and methods

## Animals

For dendritic morphology experiments, Thy1-GFP-M (Jax Stock #007788) transgenic mice (*n*=7) were used for sparse expression of GFP throughout the forebrain. For forebrain-wide calcium indicator expression, Emx1-IRES-Cre (Jax Stock #005628) × ROSA-LNL-tTA (Jax Stock #011008) × TITL-GCaMP6s (Jax Stock #024104) triple transgenic mice (*n*=2) or Slc17a7-IRES2-Cre (Jax Stock #023527) × TITL2-GC6s-ICL-TTA2 (Jax Stock #031562) double transgenic mice (*n*=6) were bred to express GCaMP6s in excitatory neurons. For imaging experiments, 8- to 51-week-old (median 17 weeks) mice of both sexes (6 males and 9 females) were implanted with a head plate and cranial window and imaged starting 2 weeks after recovery from surgical procedures and up to 10 months after microperiscope implantation. The animals were housed on a 12 hr light/dark cycle in cages of up to five animals before the implants, and individually after the implants. All animal procedures were approved by the Institutional Animal Care and Use Committee at University of California, Santa Barbara, CA.

## Surgical procedures

All surgeries were conducted under isoflurane anesthesia (3.5% induction, 1.5–2.5% maintenance). Prior to incision, the scalp was infiltrated with lidocaine (5 mg kg$^{-1}$, subcutaneous) for analgesia and meloxicam (2 mg kg$^{-1}$, subcutaneous) was administered preoperatively to reduce inflammation. Once anesthetized, the scalp overlying the dorsal skull was sanitized and removed. The periosteum was removed with a scalpel and the skull was abraded with a drill burr to improve adhesion of dental acrylic.

For hippocampal imaging, we used two types of custom-designed glass microperiscope (Tower Optical). The first (v1$_{CA1}$), for imaging the upper part of the hippocampus (CA1/CA2) consisted of a 1 × 1 × 1 mm³ square base and a 1 mm right angle prism, for a total length of 2 mm on the longest side (*Figure 1B*, left). The second (v2$_{HPC}$), for imaging the entire transverse circuit (CA1-3, DG) had a 1.5 × 1.5 × 1.0 mm³ (L × W × H) square base and a 1.5 mm right angle prism, for a total length of 2.5 mm on the longest side (*Figure 1B*, right). The hypotenuse of the right angle prisms were coated with enhanced aluminum for internal reflectance. The microperiscope was attached to a 5 mm diameter coverglass (Warner Instruments) with a UV-cured optical adhesive (Norland, NOA61). Prior to implantation, the skull was soaked in sterile saline and the cortical vasculature was inspected to ensure that no major blood vessels crossed the incision site. If the cortical vasculature was suitable, a 3–4 mm craniotomy was made over the implantation site (centered at 2.2 mm posterior, 1.2–1.7 mm lateral to Bregma). For the smaller microperiscope (v1$_{CA1}$), a 1 mm length

anterior-to-posterior incision centered at –2.1 mm posterior, 1.2 mm lateral to Bregma was then made through the dura, cortex, and mediodorsal tip of the hippocampus to a depth of 2.2 mm from the pial surface with a sterilized diamond micro knife (Fine Science Tools, #10100-30) mounted on a manipulator (*Figure 1—figure supplement 1A*). For the larger microperiscope (v2$_{HPC}$), two overlapping 1.0 mm length anterior-to-posterior incisions were made centered at –1.8 mm posterior/1.7 mm lateral and –2.4 mm posterior/1.7 mm lateral to Bregma to a depth of 2.7 mm, with a total anterior-to-posterior incision length of 1.6 mm. Note that placements in the regions shown in *Figure 1B* required incision coordinates slightly posterior to those indicated on the atlas. Care was taken not to sever any major cortical blood vessels. Gelfoam (VWR) soaked in sterile saline was used to remove any blood from the incision site. Once the incision site had no bleeding, the craniotomy was submerged in cold sterile saline, and the microperiscope was lowered into the cortex using a manipulator, with the imaging face of the microperiscope facing lateral (*Figure 1— figure supplement 1B*). Once the microperiscope assembly was completely lowered through the incision until the coverglass was flush with the skull, the edges of the window were sealed with silicon elastomer (Kwik-Sil, World Precision Instruments), then with dental acrylic (C&B-Metabond, Parkell) mixed with black ink (*Figure 1—figure supplement 1C*). Care was taken that the dental cement did not protrude over the window, as it could potentially scratch the objective lens surface. Given the working distance of the objective used in this study (3 mm), the smaller microperiscope (v1$_{CA1}$) implant enabled imaging from 2250 to 2600 μm below the coverglass surface, corresponding to approximately 150–500 μm into the lateral hippocampus (the 150 μm of tissue nearest to the microperiscope face was not used for imaging). The larger microperiscope (v2$_{HPC}$) implant enabled imaging from 2650 to 2850 μm below the coverglass surface, corresponding to approximately 150–350 μm into the lateral hippocampus (the 150 μm of tissue nearest to the microperiscope face was not used for imaging). The microperiscope implantations were stable for up to 10 months following the surgery (*Figure 4E*).

After microperiscope implantation, a custom-designed stainless steel head plate (https://www.emachineshop.com/) was affixed using dental acrylic (C&B-Metabond, Parkell) mixed with black ink. After surgery, mice were administered carprofen (5–10 mg kg$^{-1}$, oral) every 24 hr for 3 days to reduce inflammation. Microperiscope designs and head fixation hardware are available on our institutional lab web site (https://goard.mcdb.ucsb.edu/resources).

## PSF measurements

To measure empirical PSFs, fluorescent microspheres (0.2 μm yellow-green fluorescent microspheres; ThermoFisher F8811) were embedded 1:2000 in 0.5% agar and placed under the cranial window or on the face of the microperiscope. Image stacks were taken through the microspheres (0.07 μm per pixel in *XY*; 0.5 μm per plane in *Z*) located 50–100 μm from the microperiscope face (except for the microsphere-microperiscope distance comparisons in *Figure 2J–K*, which ranged from 25 to 500 μm from the microperiscope face). Candidate microspheres were initially isolated using the FindCircles function in the Image Segmenter app (MATLAB image processing toolbox). Only microspheres that were >25 pixels (1.7 μm) away from nearest neighbor microspheres and were completely contained within the *Z*-stack were used for further analysis. We registered isolated microspheres at their centroids and measured the FWHM of the average *XY* and *XZ* profiles to determine the lateral and axial resolution, respectively.

Since the geometry of the microperiscope limits the angle of the focusing light cone through the microperiscope, it predominately determines the functional numerical aperture at the imaging plane. Based on the microperiscope geometry, we calculated the effective NA of the v1$_{CA1}$ microperiscope and v2$_{HPC}$, and used it to calculate the theoretical lateral and axial resolution according to the following formulae (*Zipfel et al., 2003*):

$$\omega_{XY} = \begin{cases} \frac{0.320\,\lambda}{\sqrt{2}\,NA} & NA < 0.7 \\ \frac{0.325\,\lambda}{\sqrt{2}\,NA^{0.91}} & NA > 0.7 \end{cases}$$

$$\omega_Z = \frac{0.532\,\lambda}{\sqrt{2}} \left[ \frac{1}{n - \sqrt{n^2 - NA^2}} \right].$$

where $\omega_{XY}$ and $\omega_Z$ are the theoretical $1/e$ widths of the lateral and axial PSF, $\lambda$ is the wavelength, and *NA* is the numerical aperture. Note that the FWHM was calculated by multiplying the $1/e$ width ($\omega$) by $2\sqrt{\ln(2)}$.

To perform aberration correction with adaptive optics, a deformable mirror (Multi-3.5, Boston Micromachines Corporation) was set at a plane conjugate to the raster scanning mirrors and the back aperture of the objective lens in the 2P imaging system. Fluorescent microspheres (0.2 µm) were imaged, and the standard deviation of the image brightness was maximized under different configurations of the deformable mirror. Twelve selected Zernike modes are applied and modulated sequentially over a total of three rounds. The 12 Zernike modes are: (1) oblique astigmatism, (2) vertical astigmatism, (3) vertical trefoil, (4) vertical coma, (5) horizontal coma, (6) oblique trefoil, (7) oblique quadrafoil, (8) oblique secondary astigmatism, (9) primary spherical, (10) vertical secondary astigmatism, (11) vertical quadrafoil, (12) secondary spherical. For each Zernike mode, 21 steps of amplitudes were scanned through, and images were acquired for each step. The amplitude that resulted in the largest standard deviation was saved and set as the starting point of the DM configuration for the scanning of the next Zernike mode. The brightness, the lateral resolution, and the axial resolution are compared with and without the application of the deformable mirror correction.

## Air-floated chamber

For measurement of spatial responses, mice were head-fixed in a floating carbon fiber chamber (*Kislin et al., 2014*) (Mobile Homecage, NeuroTar, Ltd). The chamber base was embedded with magnets to allow continual tracking of the position and angular displacement of the chamber. Behavioral data was collected via the Mobile HomeCage motion tracking software (NeuroTar, versions 2.2.0.9, 2.2.014, and 2.2.1.0 beta 1). During imaging experiments, image acquisition was triggered using a TTL pulse from the behavioral software to synchronize the timestamps from the 2P imaging and chamber tracking.

A custom carbon fiber arena (250 mm diameter) was lined with four distinct visual patterns (5.7 cm tall, 18.1 cm wide) printed on 7 mil waterproof paper (TerraSlate) with black rectangles (5.7 cm tall and 1.5 cm wide) placed in between the four patterns. A circular track (*Figure 5A*; *Video 4*) was made by adding a removable inner circle (14 cm in diameter and 4.2 cm tall) with visual cues that were matched to the outer wall printed on 7 mil waterproof paper. The resulting circumference, along the middle of the circular track, is 61.26 cm. Transparent tactile stickers (Dragon Grips) were placed on the arena floors to give differential tactile stimuli along the track. In between each recording and/or behavioral session, the arena walls and floors were thoroughly cleaned.

Mice were acclimated to the arena by the following steps: (1) On the first day the mice were placed into the chamber and allowed to freely explore without head fixation for 15–20 min. A piece of plexiglass with holes drilled through was placed on top of the arena to keep the mice from climbing out. (2) On the second day, the mice were head-fixed to a crossbar extending over the floating chamber (*Figure 5A*) and allowed to freely explore the floating chamber freely for 15 min. Air flow (3–6.5 psi) was adjusted to maximize steady walking/running. On subsequent days, the head fixation time was increased by increments of 5 min, as long as the mice showed increased distance walked and percent time moving. This was continued until the mice would explore for 30–40 min and run for greater than 15% of the time. (3) Mice were head-fixed in the floated chamber for 20 min with a custom light blocker attached to their headplate. (4) Mice were head-fixed and placed on the 2P microscope to allow habituation to the microscope noise. (5) After mice were fully habituated, 20–40 min duration, recording sessions on the 2P microscope were performed.

If at any point during the above acclimation protocol the mouse significantly decreased distance traveled or percentage of time moving, then the mouse was moved back to the previous step.

Custom software was written to process the behavioral data output by the Mobile HomeCage motion tracking software. Because the Mobile HomeCage motion tracking software sampling rate was faster than the frame rate of our 2P imaging, all behavioral variables (speed, location, polar coordinates, and heading) that were captured within the acquisition of a single 2P frame were grouped together and their median value was used in future analysis. For the polar angle (which we used as the location of the mouse in 1D track), the median was computed using an open source circular statistics toolbox (CircStat 2012a) written for MATLAB (*Berens, 2009*). We removed any time points when the mouse was not moving, as is standard for measurement of place fields (*Dombeck et al., 2010*). This helps separate processes that are related to navigation from those that are related to resting state. To

do this, we smoothed the measured instantaneous speed and kept time periods > 1 s that had speeds greater than 20 mm/s (adding an additional 0.5 s buffer on either side of each time period).

## Two-photon imaging

After recovery from surgery and behavioral acclimation, GFP or GCaMP6s fluorescence was imaged using a Prairie Investigator 2P microscopy system with a resonant galvo scanning module (Bruker). For fluorescence excitation, we used a Ti:Sapphire laser (Mai-Tai eHP, Newport) with dispersion compensation (Deep See, Newport) tuned to $\lambda$=920 nm. Laser power ranged from 40 to 80 mW at the sample depending on GCaMP6s expression levels. Photobleaching was minimal (<1% min$^{-1}$) for all laser powers used. For collection, we used GaAsP photomultiplier tubes (H10770PA-40, Hamamatsu). A custom stainless-steel light blocker (https://www.emachineshop.com/start/) was mounted to the head plate and interlocked with a tube around the objective to prevent light from the environment from reaching the photomultiplier tubes. For imaging, we used a 16×/0.8 NA microscope objective (Nikon) to collect 760 × 760 pixel frames with field sizes of 829 × 829 or 415 × 415 µm$^2$. Images were collected at 20 Hz and stored at 10 Hz, averaging two scans for each image to reduce shot noise.

For imaging spines across days, imaging fields on a given recording session were aligned based on the average projection from a reference session, guided by stable structural landmarks such as specific neurons and dendrites. Physical controls were used to ensure precise placement of the head plate, and data acquisition settings were kept consistent across sessions. Images were collected once every day for 5–10 days.

## Two-photon post-processing

Images were acquired using PrairieView acquisition software (Bruker) and converted into multi-page TIF files.

For spine imaging, registration and averaging was performed for each *z*-plane spanning the axial width of the dendrite to ensure all spines were captured across *z*-planes. The resulting projections were weighted according to a Gaussian distribution across planes. Non-rigid registration was used to align dendritic segments across consecutive recording sessions. The registered images underwent high-pass filtering to extract low amplitude spine features using code adapted from Suite2P's enhanced mean image function (*Pachitariu et al., 2016*) (*Figure 3—figure supplement 1A*). The resulting ROIs were binarized using Otsu's global threshold method for spine classification (*Figure 3—figure supplement 1A*). In most cases, the global threshold successfully isolated the single most prominent dendrite. In fields with higher background dendrites that were not desired, these extraneous dendrites were manually excluded. To identify spines that fall below the global threshold, the user manually specifies incrementally lower thresholds from which to select spines that were excluded in the initial binarization. (*Figure 3—figure supplement 1A*). Spines above the global threshold with an area of >1 µm$^2$ were included in our analysis. To classify each spine as one of the four major morphological classes, we performed the following steps. First, we found the base of the spine by identifying the region closest to the dendritic shaft. Second, we calculated the length of the spine by taking the Euclidean distance between the midpoint of the spine base and the most distant pixel. Third, this vector was divided evenly into three segments to find the spine head, neck, and base areas, respectively. Finally, spines were classified in the four categories, considering the following threshold parameters (*Figure 3—figure supplement 1B*): stubby (neck length <0.2 µm and aspect ratio <1.3), thin (neck length >0.2 µm, spine length <0.7 µm, head circularity <0.8 µm), mushroom (neck length >0.2 µm, head circularity >0.8 µm), and filopodium (neck length >0.2 µm, spine length <0.8 µm, aspect ratio >1.3).

For calcium imaging sessions, the TIF files were processed using the Python implementation of Suite2P (*Pachitariu et al., 2016*). We briefly summarize their pipeline here. First, TIFs in the image stack undergo rigid registration using regularized phase correlations. This involves spatial whitening and then cross-correlating frames. Next, regions of interest (ROIs) are extracted by clustering correlated pixels, where a low-dimensional decomposition is used to reduce the size of the data. The number of ROIs is set automatically from a threshold set on the pixel correlations. We manually checked assigned ROIs based on location, morphology, and *DF/F* traces.

Since the hippocampal pyramidal cells are densely packed and the microperiscope reduces the axial resolution, we perform local neuropil subtraction using custom code (https://github.com/

ucsb-goard-lab/two-photon-calcium-post-processing; *Kait, 2022*) to avoid neuropil contamination. The corrected fluorescence was estimated according to

$$F_{corrected}(n) = F_{some}(n) - \alpha \, (F_{neurophil}(n) - \bar{F}_{neurophil})$$

where $F_{\text{neuropil}}$ was defined as the fluorescence in the region <30 μm from the ROI border (excluding other ROIs) for frame $n$. $\bar{F}_{\text{neuropil}}$ is $F_{\text{neuropil}}$ averaged over all frames. $\alpha$ was chosen from [0, 1] to minimize the Pearson's correlation coefficient between $F_{\text{corrected}}$ and $F_{\text{neuropil}}$. The ΔF/F for each neuron was then calculated as

$$\frac{\Delta F}{F} = \frac{F_n - F_0}{F_0},$$

where $F_n$ is the corrected fluorescence ($F_{corrected}$) for frame $n$ and $F_0$ is defined as the first mode of the corrected fluorescence density distribution across the entire time series.

We deconvolved this neuropil subtracted ΔF/F to obtain an estimate for the instantaneous spike rate, which we used (only) for the computation of neurons' spatial information (see below). This inferred spike rate was obtained via a MATLAB implementation of a sparse, nonnegative deconvolution algorithm (OASIS) used for $Ca^{2+}$ recordings (*Friedrich et al., 2017*). We used an auto-regressive model of order 2 for the convolution kernel. Any cells that had spike rate >10 spikes/s or <1 spikes/s were manually checked and were removed from consideration if their traces appeared too noisy or sparse. Such cells were not considered in future analysis and were not included in the total number of cells recorded from.

## Spine imaging data analysis

After non-rigid registration, high-pass filtering, and binarization of the dendritic segment, individual spines were extracted based on standard morphological criteria (*Holtmaat and Svoboda, 2009*). Spines projecting laterally from the dendritic segment were extracted and analyzed as individual objects, as described previously (*Figure 3—figure supplement 1*). The sum of the members of each spine class, as well as the total number of all spines, was recorded for each session. Spine totals ($S_{total}$) were then broken down into 10 μm sections of the dendritic segment ($S_{section}$) using the following calculation

$$S_{section}(n) = \frac{S_{total}(n)}{\left(D_{length} \times \frac{F_{\mu m}}{F_{pixels}}\right)} \times 10$$

where length of the dendritic segment, $D_{length}$, was determined by skeletonizing the dendritic shaft to 1 pixel in diameter, then taking the area of the pixels. $F_{pixels}$ is the FOV in pixels, which here was 760 × 760 at 16× magnification, and $F_{\mu m}$ is the FOV in μm, which was 52 × 52 μm². 

Turnover was estimated at 24 hr increments; turnover here is defined as the net change in spines per day for each morphological class (*Figure 3F*). To determine which specific spines were involved in turnover across days, segments recorded 24 hr apart were aligned and overlaid using a custom MATLAB interface, which allowed the user to manually select new or removed spines. Percent addition/subtraction $S_{a/s}$ was calculated as

$$S_{a/s} = \frac{N_{a/s}(t)}{N(t)} \times 100,$$

where $N_{a/s}(t)$ is spines that have been added or subtracted and $N(t)$ is the total average number of spines. To account for variance in spine classification across days, turnover of specific classes of spines was normalized to total cumulative turnover per day.

To calculate the survival fraction curve $S(t)$, we determined which spines were present at time $t_n$ that were not present at time $t_0$ (*Attardo et al., 2015*; *Mizrahi et al., 2004*; *Pfeiffer et al., 2018*). The dendritic segment from $t_0$ was transparently overlaid with segments from $t_n$, and replacement spines that were present in $t_0$ but not $t_n$ were manually identified. Survival fraction was quantified as

$$S(t) = \frac{N_r(t_n)}{N(t_0)} \times 100$$

where $N_r(t_n)$ are the total spines at $t_n$ that were also found in $t_0$, and $N(t_0)$ are the total number of spines that were present in $t_0$. Survival fraction, as well as % addition and subtraction, was calculated in 10 µm sections to control for segment length.

Dendrite registration (*Figure 3—figure supplement 2*) was performed using built-in registration functions in the MATLAB image processing toolbox (rigid registration: imregister, non-rigid registration: imregdemons for displacement field estimation and warping).

## Calcium imaging data analysis

For calcium imaging experiments during exploration of the air-floated chamber, processed and synchronized behavioral data and 2P imaging data were used to identify PCs and SCs as follows.

First, the 1D track was divided into 72 equal bins (each ~0.85 cm in length). Activity as a function of position (we refer to these as spatial tuning curves) was computed for each lap, with activity divided by occupancy of each binned location. We observed that in certain cases, the mice traversed the track at high speeds. To avoid misattribution of slow calcium signals to spatial bins (which were relatively small due to our small track), any lap where the average instantaneous speed was greater than 180 mm/s was removed and not considered for further analysis (an average of 7% of laps were removed). To assess the consistency of spatial coding of each cell, we randomly split the laps into two groups and computed the correlation coefficient between the averaged spatial tuning curves. We then did the same for shuffled data in which each lap's spatial tuning curve was circularly permuted by a random number of bins. Note that this was done for each lap, to avoid trivial effects that might emerge from circularly permuting data that was stereotyped along the track. This was performed 500 times, and the distribution of actual correlation coefficient values was compared to the distribution of circularly shuffled values using a two-sample Kolmogorov-Smirnov test ($\alpha$=0.01). The distribution also had to pass a Cohen's $D$ analysis, having a score of greater than 0.5. A cell that passed these tests was considered a 'consistent' cell.

To identify a neuron as a PC, the neuron had to pass the consistency test, in addition to being well fit by a Gaussian function, $R_{DF/F} = A_0 + A \, e^{\left(\frac{(X-B)}{C}\right)^2}$, with FWHM = $2C\sqrt{\ln 2}$. Note that in this convention, $C^2 = 2\sigma^2$. Specifically, we required that: (1) the adjusted $R^2$ >0.375; (2) 2.5 cm <FWHM < 30.6 cm (50% of track length – results in *Figure 5H* did not significantly depend on this threshold); (3) $A$>0; (4) $A/A_0$>0.50. Cells that met these conditions were characterized as PCs; with place fields at the location of maximal activity and width defined as the FWHM. Note that these criteria are somewhat strict compared to traditional place field criteria. When tested with data in which individual laps were time shuffled, the approach yielded a false positive rate of 0%.

SCs were identified using a standard process developed for identification of SCs in medial EC and hippocampus (*Kropff et al., 2015*; *Iwase et al., 2020*; *Góis and Tort, 2018*). We computed the Pearson's correlation of each cells' $DF/F$ trace with the mouse's speed across the experiment. This value is considered as a 'speed score'. We then circularly shuffled the $DF/F$ 100 times (making sure that the amount shuffled was greater than 10 frames to ensure that the shuffled distribution did not have artificially high correlations). Cells whose speed score was greater than 99%, or less than 1%, of the shuffled distribution were considered SCs.

To compute the spatial information (*Skaggs et al., 1993*) of cell $j$ ($SI_j$), we used the following formula,

$$SI_j = \frac{1}{\bar{a}_j} \sum_{k=1}^{72} p(k) a_j(k) \, log_2 \left[ \frac{a_j(k)}{\bar{a}_j} \right]$$

where $\bar{a}_j$ is the mean inferred spike rate of cell $j$, $a_j(k)$ is the mean inferred spike rate of cell $j$ at position bin $k$, and $p(k)$ is the probability of being at position bin $k$. We divide by $\bar{a}_j$ to have $SI$ in units of bits/inferred spike. To align recordings where we recorded along the CA1-DG axis, we found the inflection point of the axis and then computed the distance of each cell to that point. To do this, we performed the following steps. (1) We extracted the position of each identified cell using Suite2P's centroid output. (2) We then fit a function of the form $a(x-b)^2 + c$ to the cell positions by rotating the field-of-view from 0 to 180 degrees and finding the rotation that maximized the $R^2$ value of the fit. (3) We determined the inflection point as the peak of the curve and de-rotated the fit to determine the

inflection point and curve in the original coordinates. The distance of each cell to the inflection point was found by finding the point along the fit curve that had the minimal distance to the cell's centroid.

## Immunohistochemistry

Samples were perfusion fixed using 4% paraformaldehyde in 0.1 M sodium cacodylate buffer (pH = 7.4) for 10 min, and then immersion fixed overnight at 4°C. Next, sections were rinsed in cold PBS 5 × 5 min and 1 × 1 hr. Whole brains were then embedded in 10% low-melting agarose. Subsequently, 100 μm coronal sections were cut using a vibratome (Leica, Lumberton, NJ). Sections were then blocked overnight in normal donkey serum (Jackson ImmunoResearch; West Grove, PA) diluted 1:20 in PBS containing 0.5% bovine serum albumin, 0.1% Triton X-100, and 0.1% sodium azide, hereafter, PBTA at 4°C. Next, primary antibodies anti-GFAP (1:500; abcam; ab53554), anti-S100β (1:1000; DAKO; Z0311) were diluted in PBTA and incubated overnight at 4°C. Then, sections were rinsed 5 × 5 min and 1 × 1 hr before corresponding secondary antibodies along with the nuclear stain Hoechst33342 (1:5000; Molecular Probes; H-3570) were incubated overnight at 4°C. Lastly, secondary antibodies were rinsed, and sections mounted using Vectashield (Vector laboratories Inc; H-1200) and sealed under #0 coverslips.

High-resolution wide-field mosaics of brain samples were then imaged with a 20× oil immersion lens and an Olympus Fluoview 1000 laser scanning confocal microscope (Center Valley, PA) at a pixel array of 800 × 800 and then registered using the bio-image software Imago (Santa Barbara, CA).

We then calculated glial cell density as a function of distance from the microperiscope face. First, each mosaic was rotated so that the medial-lateral axis of the brain sample was aligned to be parallel with the horizontal axis of the image. Then, each mosaic was cropped to remove extraneous pixels outside of the imaged brain slice. Next, a line denoting the face of the microperiscope was manually drawn parallel to the dorsal-ventral axis aligned with the location of the microperiscope face. We then used a custom cell-counting algorithm that identified potential ROIs. We limited the ROIs to be within the hippocampal formation in the brain slices. The Euclidean distance between the closest point on the defined microperiscope face and each ROI's centroid was calculated. Afterward, a similar procedure was performed on the contralateral side of the brain slice, with a mock 'microperiscope face' defined at symmetric coordinates to the true microperiscope face, to serve as a control. These steps were repeated for each channel of the mosaic.

After extracting each ROIs distance from the microperiscope face, we counted the number of cells in each 50 μm distance bin. To account for basal glial cell density, we calculated the percent change of glial cell density on the microperiscope side with respect to the control side. This procedure was repeated 1000 times using randomly sampled distances, with replacement, to bootstrap the sample variance.

## Statistical information

Violin plots were made using an open-source MATLAB package (*Bechtold, 2016Bechtold, 2016*). Statistical tests for spine morphological types were calculated using a one-way ANOVA. Reliability across laps was tested with a two-sample Kolmogorov-Smirnov test. Comparisons between model fits for spatial distribution of spatial information and place field width used a general linear F-test.

## Acknowledgements

We would like to thank Caleb Kamere and Eliott Levy for comments on the manuscript. This work was supported by NSF (MJG, SLS, NeuroNex #1707287), NIH (MJG, SLS, R01 RF1NS121919), the Larry Hillblom foundation (MJG), and the Whitehall Foundation (MJG).

## Additional information

### Funding

| Funder | Grant reference number | Author |
|---|---|---|
| National Science Foundation | 1707287 | Spencer Smith Michael J Goard |
| National Institute of Neurological Disorders and Stroke | RF1NS121919 | Spencer Smith Michael J Goard |
| Whitehall Foundation | | Michael J Goard |
| Larry L. Hillblom Foundation | | Michael J Goard |

The funders had no role in study design, data collection and interpretation, or the decision to submit the work for publication.

### Author contributions

William T Redman, Conceptualization, Formal analysis, Investigation, Software, Visualization, Writing – original draft, Writing – review and editing; Nora S Wolcott, Formal analysis, Investigation, Software, Visualization, Writing – original draft, Writing – review and editing; Luca Montelisciani, Formal analysis, Investigation, Software, Visualization; Gabriel Luna, Tyler D Marks, Investigation; Kevin K Sit, Software, Visualization; Che-Hang Yu, Formal analysis, Methodology; Spencer Smith, Funding acquisition, Methodology, Project administration, Writing – review and editing; Michael J Goard, Conceptualization, Formal analysis, Funding acquisition, Investigation, Methodology, Project administration, Supervision, Surgical procedures, Visualization, Writing – original draft, Writing – review and editing

### Author ORCIDs

William T Redman ⓘ http://orcid.org/0000-0002-4147-2026
Che-Hang Yu ⓘ http://orcid.org/0000-0002-0353-9752
Michael J Goard ⓘ http://orcid.org/0000-0002-5366-8501

### Ethics

All animal procedures were approved by the Institutional Animal Care and Use Committee at University of California, Santa Barbara (Protocol #906.1).

### Decision letter and Author response

Decision letter https://doi.org/10.7554/eLife.75391.sa1
Author response https://doi.org/10.7554/eLife.75391.sa2

## Additional files

### Supplementary files
• Transparent reporting form

### Data availability

Microperiscope designs are available on Dryad (https://doi.org/10.25349/D9T903). Microsphere data from Figure 2, spine imaging data from Figure 3, and neuronal response data from Figures 5-6 are available on Dryad (https://doi.org/10.25349/D9T903). Code for point spread function calculations, spine analysis, and place/speed cell identification is available on Github (https://github.com/ucsb-goard-lab/HippocampusMicroperiscope, copy archived at swh:1:rev:e92c248a37623d571c3fc730a821888243fdf20c).

The following dataset was generated:

| Author(s) | Year | Dataset title | Dataset URL | Database and Identifier |
|---|---|---|---|---|
| Redman WT, Wolcott N, Montelisciani L, Luna G, Marks T, Sit K, Yu C, Smith S, Goard M | 2022 | Long-term Transverse Imaging of the Hippocampus with Glass Microperiscopes | https://doi.org/10.25349/D9T903 | Dryad Digital Repository, 10.25349/D9T903 |

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
