## [Editor Report]

This paper presents new optical technologies that allow the investigation of all stages of the tri-synaptic hippocampal circuit during behavior within an individual animal. The approach is a major methodological advance and the authors use it to confirm differences in spatial selectivity of place cells across hippocampal subregions. The paper will be of interest to the large number of neuroscientists who study the hippocampal circuit, and more broadly to those interested in methods to enable high-resolution in vivo imaging across multiple depths in the brain.

---

## [Decision Letter]

**Decision letter after peer review:**

Thank you for submitting your article "Long-term Transverse Imaging of the Hippocampus with Glass Microperiscopes" for consideration by *eLife*. Your article has been reviewed by 3 peer reviewers, and the evaluation has been overseen by a Reviewing Editor and Laura Colgin as the Senior Editor. The following individuals involved in review of your submission have agreed to reveal their identity: Mark Sheffield (Reviewer #2); Jim Heys (Reviewer #3).

All three reviewers, as you will see below, expressed enthusiasm regarding the manuscript and the approach described. However, there are a few essential suggested revisions that should be incorporated prior to acceptance of the manuscript at *eLife*, as listed below.

1) Reviewer 1 provides a number of suggestions (see recommendations for authors) requiring the addition of a figure panel to show actual GFAP+ cells, a panel showing the full extent of the implant, minor text changes, and an additional figure panel to present the results of the modeling.

2) Reviewer 3, and all three reviewers in discussion, agreed that additional analyses were needed regarding the long-term health of the brain tissue near the microperiscope implantation. This could include measures of place cells (field stability) or % of overfilled cells, as a function of distance from the prism face for FOVs across the three hippocampal subfields (see detailed comment from reviewer 3, comment 1), or measures of glial markers across the duration of imaging. In the case that analyses are not sufficient to determine the long-term health of brain tissue, discussion regarding this as a potential caveat of the method should be added.

3) Additional analyses would also improve the clarity of the results, as outlined in Reviewer 3, comment 2 (see recommendations for authors). These include comments which require minor text changes (comments 2b and 2d), a suggested figure panel (comment 2a) and a minor analysis (comment 2c).

4) Reviewer 3 has several minor comments that should be addressed (e.g. typos, citations, clarifications).

*Reviewer #1 (Recommendations for the authors):*

The characterization of the impact of implanting the micro periscopes in the hippocampus could be strengthened by a panel showing actual GFAP+ cells and a panel showing the full extent of the implant instead or in addition to Figure 1E.

The optical characterization of the micro periscopes is very clear, and the figure 2 is visually pleasing, but I think there are a few points the authors should address.

1 – From lines 102 to 106 the authors comment on the clipping of the excitation beam. This is an important issue that they investigated using modelling. If I were to use the micro periscopes I'd really like to know how "robust" my imaging will be as I go further from the surface of the prism. The authors should share the results of the modelling in a figure or, alternatively, they could repeat the measurement in Figure 2 D to G at a further distance from the micro periscope.

2 – On the same issue, I could not find the value of the distance from the surface of the beads imaged in Figure 2.

3 – In lines 99 to 100 the authors, state that a FWHM of 0.7 is "similar" to a FWHM of 1.0. This statement seem at odds with the strong quantitative attitude of the authors. How are two numbers differing by ~30% "similar"?

Regarding the structural imaging, it is evident that authors can detect and track dendritic spines in the apical dorsal CA1. However, I see a few overstatements the authors should revise.

1 – Line 118, the authors do not show the extent to which they can actually resolve spines. They can detect spines, but it is unclear to what extent they can detect all the spines, or otherwise what is the minimum distance at which they can resolve two neighboring spines. This is important when comparing their work to previous work (namely Attardo and Pfeiffer).

2 – The statement in line 124 ("imaging the apical dendritic spines of hippocampal neurons has not previously been possible") is not correct in that both Schmid et al., Neuron (2016) and Ulivi et al., JoVE (2019) showed imaging of dendritic spines in the apical aspect of dorsal CA1 (Figure 3 and Figure 2 respectively). This applies also to the statement on lines 287 and 288.

3 – As detailed in the public review, the authors should expand the discussion of the interpretation of their results in the light of previous literature on stability of spines in dorsal CA1.

4 – I think it would be appropriate to cite Gu et al., J. Neurosci. (2014) – on how underestimation of the limitations in resolution might confound the interpretation of spine dynamics – and Castello-Waldow et al., PLoS Biology (2020) – on a how to control for the limitations in resolution while. Both papers deal with tracking dendritic spines in pyramidal neurons of the dorsal CA1. For historical completeness I'd also cite Mizrahi et al., J. Neurosci. (2014) as the first example of in vivo tracking of dendritic spines in the dorsal CA1.

*Reviewer #2 (Recommendations for the authors):*

This paper is very well written and figures are clear and easy to read. The conclusions made are supported by the data and the methods developed are an important advance for the field. I personally look forward to seeing how this method will be used in the future to provide new insights into Hippocampal function.

I have no concerns to be addressed.

*Reviewer #3 (Recommendations for the authors):*

The authors should be commended on their development of this imaging approach, which could greatly advance the field's understanding of the hippocampal circuit. A key strength of this approach is to simultaneously monitor hippocampal dynamics across CA1, CA3 and DG subfields. Although prior work has investigated these questions across hippocampal subfields, recordings across subfields has been done primarily across different animals. In some cases simultaneous recordings have been across subfields, but the throughput is low and it is difficult to gain clear population-level picture of the neural dynamics across regions. It is well known within this field that phenomenon like remapping can be very subtle and sensitive to contextual details, and also very different across subjects. Therefore, in order to accurately assess how these regions differentially contribute to memory it is necessary to measure neural representations simultaneously across these regions within the same animal. This makes the methodological advancement in this manuscript compelling. As alluded to in the public review, I have a couple major critiques and several minor suggestions which could aid interpretation of this manuscript.

1. As this manuscript exists, it is difficult to assess the effect of microperiscope implantation on the brain. While glial markers are noted, it is unclear when these data were collected. Additionally, given that authors image for months, it would be beneficial to see longitudinal changes in glial markers through the duration of imaging. Additionally, authors are missing a key opportunity to evaluate the effect of the implant by measuring place cell properties as a function of distance from the microperiscope. I expect cells nearby may be unhealthy and show differences in place field width or simply the average transient duration. The authors should characterize place field width, place field stability, transient on/off time for smaller transients (i.e. those caused by 1 or 2 action potentials), and % of overfilled cells, all measured as a function of distance from the prism face for FOVs across the 3 HPC subfields. This could help determine whether any differences in spatial information across the CA1-DG axis are caused by differential implant or cell health dependent vulnerabilities across these regions.

2. In the absence of more detailed place cell characterization, it is difficult to evaluate how well this imaging and behavioral approach captures place field dynamics across hippocampal subfields.

a. Please show examples of trial-by-trial raster data for place cells from CA1,CA3, and DG.

b. It is noted that laps where the animal ran faster than 1s were removed, as were periods of running >18 cm/s. This is not very fast relative to average mouse velocity on a linear or circular track. Please describe quantitatively how much data was removed due to these constraints.

c. The constraints on place field width (<30cm) seem to cut off part of the distribution. Could this be accounting for the NS effect seen across CA1-DG in Figure 5G? Additionally, place field width is determined by the spatial selectivity of firing across the track, which will vary across laps. The authors should include a measure of COM across laps as well as average field width/lap to determine differences in place field width that emerge when taking these distinct aspects of place fields into account.

d. The behavioral paradigm does not include a reward, while nearly all prior recordings of hippocampal place cells are in the context of appetitive instrumental tasks. The effect of removing reward from a task can have a major influence on stability (see preprint for Krishnan et al. Biorxiv). Perhaps authors could comment on whether the difference in reward should guide interpretation of findings.

---

## [Author Response]

Reviewer #1 (Recommendations for the authors):The characterization of the impact of implanting the micro periscopes in the hippocampus could be strengthened by a panel showing actual GFAP+ cells and a panel showing the full extent of the implant instead or in addition to Figure 1E.

Agreed. In Figure 1E, we wanted to show a higher magnification view so that readers could see the pattern of astrocyte labeling near the microperiscope face, but we added a new supplementary figure showing the histology at a range of magnifications, from the entire slice to cellular-level labeling (Figure S2).

The optical characterization of the micro periscopes is very clear, and the figure 2 is visually pleasing, but I think there are a few points the authors should address.1 – From lines 102 to 106 the authors comment on the clipping of the excitation beam. This is an important issue that they investigated using modelling. If I were to use the micro periscopes I'd really like to know how "robust" my imaging will be as I go further from the surface of the prism. The authors should share the results of the modelling in a figure or, alternatively, they could repeat the measurement in Figure 2 D to G at a further distance from the micro periscope.

We agree that it is possible that imaging depth (in relation to the microperiscope imaging face) could theoretically affect resolution. To test this empirically, we performed fluorescent microsphere measurements in which we measured lateral and axial resolution as a function of distance from the microperiscope plane surface (Figure 2J). Although there was no discernible effect on the lateral resolution, the axial resolution appeared slightly poorer at distances >350 microns from the microperiscope face (Figure 2K; lines 111-117). We generally did not image neurons this far from the microperiscope face, as less efficient excitation and decreased signal collection efficiency led to dim fields and increased the required laser power.

2 – On the same issue, I could not find the value of the distance from the surface of the beads imaged in Figure 2.

For these measurements, the beads were measured approximately 50-100 microns from the periscope face; this has been added to the results (lines 463-466).

3 – In lines 99 to 100 the authors, state that a FWHM of 0.7 is "similar" to a FWHM of 1.0. This statement seem at odds with the strong quantitative attitude of the authors. How are two numbers differing by ~30% "similar"?

Agreed, we have changed the text to more accurately reflect the data (lines 98-101).

Regarding the structural imaging, it is evident that authors can detect and track dendritic spines in the apical dorsal CA1. However, I see a few overstatements the authors should revise.1 – Line 118, the authors do not show the extent to which they can actually resolve spines. They can detect spines, but it is unclear to what extent they can detect all the spines, or otherwise what is the minimum distance at which they can resolve two neighboring spines. This is important when comparing their work to previous work (namely Attardo and Pfeiffer).

We agree, and suspect we are missing small structures (e.g., filipodia) and densely-packed spines. In principle, super-resolution 2-photon imaging (Pfeiffer et al., 2018) should be fully compatible with the microperiscope imaging approach. Although we are not currently set up to carry out these experiments, we hope our group or others will be able to directly compare to super-resolution spine imaging in future experiments. We have addressed this point in the discussion (lines 335-340).

2 – The statement in line 124 ("imaging the apical dendritic spines of hippocampal neurons has not previously been possible") is not correct in that both Schmid et al., Neuron (2016) and Ulivi et al., JoVE (2019) showed imaging of dendritic spines in the apical aspect of dorsal CA1 (Figure 3 and Figure 2 respectively). This applies also to the statement on lines 287 and 288.

Unless we are mistaken, it appears that in Figure 3 of the Schmid et al. paper, they imaged basal dendrites in the stratum oriens, not apical dendrites. However, the reviewer is correct that Ulivi et al. imaged spines on offshoots from apical dendrites, as well as another methods paper (Gu et al., 2014). We have corrected this in the text (lines 127-130, 324-328) to reflect that previous work has only been able to image transverse offshoots of the apical dendrites, and not the major axis of the apical dendrite.

3 – As detailed in the public review, the authors should expand the discussion of the interpretation of their results in the light of previous literature on stability of spines in dorsal CA1.

Agreed. We have addressed this in the revised text as described in the public review (lines 162-165, 331-332).

4 – I think it would be appropriate to cite Gu et al., J. Neurosci. (2014) – on how underestimation of the limitations in resolution might confound the interpretation of spine dynamics – and Castello-Waldow et al., PLoS Biology (2020) – on a how to control for the limitations in resolution while. Both papers deal with tracking dendritic spines in pyramidal neurons of the dorsal CA1. For historical completeness I'd also cite Mizrahi et al., J. Neurosci. (2014) as the first example of in vivo tracking of dendritic spines in the dorsal CA1.

These citations have been added; thank you for pointing out those papers.

Reviewer #3 (Recommendations for the authors):The authors should be commended on their development of this imaging approach, which could greatly advance the field's understanding of the hippocampal circuit. A key strength of this approach is to simultaneously monitor hippocampal dynamics across CA1, CA3 and DG subfields. Although prior work has investigated these questions across hippocampal subfields, recordings across subfields has been done primarily across different animals. In some cases simultaneous recordings have been across subfields, but the throughput is low and it is difficult to gain clear population-level picture of the neural dynamics across regions. It is well known within this field that phenomenon like remapping can be very subtle and sensitive to contextual details, and also very different across subjects. Therefore, in order to accurately assess how these regions differentially contribute to memory it is necessary to measure neural representations simultaneously across these regions within the same animal. This makes the methodological advancement in this manuscript compelling. As alluded to in the public review, I have a couple major critiques and several minor suggestions which could aid interpretation of this manuscript.1. As this manuscript exists, it is difficult to assess the effect of microperiscope implantation on the brain. While glial markers are noted, it is unclear when these data were collected. Additionally, given that authors image for months, it would be beneficial to see longitudinal changes in glial markers through the duration of imaging. Additionally, authors are missing a key opportunity to evaluate the effect of the implant by measuring place cell properties as a function of distance from the microperiscope. I expect cells nearby may be unhealthy and show differences in place field width or simply the average transient duration. The authors should characterize place field width, place field stability, transient on/off time for smaller transients (i.e. those caused by 1 or 2 action potentials), and % of overfilled cells, all measured as a function of distance from the prism face for FOVs across the 3 HPC subfields. This could help determine whether any differences in spatial information across the CA1-DG axis are caused by differential implant or cell health dependent vulnerabilities across these regions.

We have added post-implant dates for histology data (Figure 1F caption). Although we agree that longitudinal tracking of glial markers would be useful, we were not able to perform those experiments without using considerably more mice (there is some variability in the immunohistochemistry, so we would need multiple mice per time point).

We thank the reviewer for the suggestion of examining functional properties as a function of distance from the face of the microperiscope. While our objective’s working distance limits a full examination of this across all subregions (particularly for the v2HPC microperiscope, in which our objective is quite close to the top of the microperiscope), we investigated the functional properties in four mice across five distances from the periscope face (30um, 80um, 130um, 180um, 230um) in CA1 and mCA3. Place cell width did not vary significantly as a function of depth of the imaging plane from the face of the periscope (Figure S7A). Spatial information varied significantly, but there was no systematic effect (Figure S7B). The decay constant for the fitted transients did have a significant relationship with periscope depth (Figure S7C), which may be due to aberrant activity in a subset of neurons close to the microperiscope face. We also found that spatial precision (Sheffield and Dombeck, 2015) did not vary significantly as a function of depth (see Author response image 1; *F*(4, 241) = 2.15, *p* = 0.08; one-way ANOVA). We did not observe clearly overfilled cells at any depth (see, for example, Video S1). We have made the results the subject of their own supplemental figure (Figure S7) and have commented on them in the main text (lines 220-226). Although spatially responsive neurons could be found at all distances, our general advice would be to only image >150 μm from the microperiscope face to ensure cell health.

**Author response image 1. sa2fig1:** Spatial precision does not vary significantly with depth of imaging plane from the face of the periscope.

2. In the absence of more detailed place cell characterization, it is difficult to evaluate how well this imaging and behavioral approach captures place field dynamics across hippocampal subfields.a. Please show examples of trial-by-trial raster data for place cells from CA1,CA3, and DG.b. It is noted that laps where the animal ran faster than 1s were removed, as were periods of running >18 cm/s. This is not very fast relative to average mouse velocity on a linear or circular track. Please describe quantitatively how much data was removed due to these constraints.c. The constraints on place field width (<30cm) seem to cut off part of the distribution. Could this be accounting for the NS effect seen across CA1-DG in Figure 5G? Additionally, place field width is determined by the spatial selectivity of firing across the track, which will vary across laps. The authors should include a measure of COM across laps as well as average field width/lap to determine differences in place field width that emerge when taking these distinct aspects of place fields into account.d. The behavioral paradigm does not include a reward, while nearly all prior recordings of hippocampal place cells are in the context of appetitive instrumental tasks. The effect of removing reward from a task can have a major influence on stability (see preprint for Krishnan et al. Biorxiv). Perhaps authors could comment on whether the difference in reward should guide interpretation of findings.

We agree with the reviewer that more details surrounding our place cell characterization would clarify the ability of our imaging and behavioral approach to capture place cell dynamics, as well as make some of the choices we made with respect to data post-processing appear less arbitrary. We address the reviewer’s points in the following ways:

a) We have added more detail about our post-processing method in the main text and in the Methods (lines 207-209, 663-666, 674-676). We have also added lap-by-lap activity and Gaussian fit of example place cells from each subregion (Figure 5C). The reason we developed our own approach is that we found that existing analysis approaches were overly inclusive, and we found a non-trivial percentage of place cells even in our shuffled data. We decided to further increase the stringency of our criteria in the revised manuscript to eliminate false positives. This reduced the percentage of place cells we found, particularly in DG (see Figure 5G), but increased our confidence that our neurons exhibit spatially-specific firing (our current approach has a 0% false positive rate in shuffled data). In terms of face validity, this approach appeared to perform well at identifying reliably spatially-modulated cells while excluding neurons that were only active on a small number of laps. As discussed in point 2d below, it should be noted that the place cells were not as reliable as those recorded in the presence of reward using a similar behavioral apparatus (Go et al., 2021), likely due to the lack of reward.

b) We realized that the choice of running speed threshold was not explained well. Because our track was small, relative to other physical and virtual environments that have been used in prior hippocampal work, the mice were able to quickly traverse the track, even at speeds that would not be considered “fast” in other experimental set-ups. Additionally, our 2-photon acquisition rate was 10Hz, which results in poor sampling at high speeds. We placed an upper bound on the average speed of a lap (note: we did not place an upper bound on the instantaneous speed) to remove such laps from contaminating our place cell analysis. Across recordings, we removed an average of 7% of the total number of laps. We have added detail to the Methods (lines 654-655, 658, 661-663) to clarify this point.

c) We specifically chose this cut-off on place field width as it was half the length of our arena. We admit that the choice of ~30 cm is, in general, on the smaller side (as again, our circular track was on the smaller side compared to other virtual and physical environments), but we wanted to remove any non-spatial cells that might show elevated activity across the majority of the track. However, we agree with the reviewer that this upper bound could be hiding a difference between the different subregions. We therefore re-did the analysis for the upper bound being 75% of the track (p(CA1, CA3) = 0.07; p(CA1, DG) = 0.65; p(CA3, DG) = 0.91; Two-sample KolmogorovSmirnov test) and for the upper bound being 100% of the track (p(CA1, CA3) = 0.16; p(CA1, DG) = 0.44; p(CA3, DG) = 0.70; Two-sample Kolmogorov-Smirnov test). In each case, the distributions across the subregions were not significantly different. We added additional discussion of this (lines 671-672) to make this point more explicit.

We performed center-of-mass (COM) analysis and found no statistically significant difference between the COM standard deviation across regions (p(CA1, CA3) = 0.59; p(CA1, DG) = 0.22; p(CA3, DG) = 0.68; Two-sample Kolmogorov-Smirnov test). Similarly, we measured the width of place fields for each lap and found no statistically significant difference between the regions (p(CA1, CA3) = 0.50; p(CA1, DG) = 0.06; p(CA3, DG) = 0.26; Two-sample Kolmogorov-Smirnov test).

d) We agree this is an important point that needs addressing. Because place cells are known to be modulated by reward (e.g. Gauthier and Tank, 2018) and we were hoping to focus on spatially specific responses, we chose to measure hippocampal responses in the absence of reward. Recent work (Krishnan et al., bioRxiv; Pettit et al., 2022), has shown that reward significantly modulates the reliability of place cell firing, which may explain why our place cells are somewhat less reliable than previous reports using a similar behavioral apparatus (Go et al., 2021). However, even without the reward, we still had a significant fraction of place cells (similar to those found in other mouse hippocampal imaging studies, though fewer than in rats) when using a strict criterion for significance (see response to point 2a. above).

We agree with the reviewer that this experimental choice is an important distinction, as it is possible that place field quantities we measured would be different in the presence of reward. We have added additional discussion of this caveat to the discussion (lines 307-318).